# SUBGRAPH-TO-NODE TRANSLATION FOR EFFICIENT REPRESENTATION LEARNING OF SUBGRAPHS

## ABSTRACT

Subgraph representation learning has emerged as an important problem, but it is by default approached with the graph neural networks (GNNs) on a large global graph, an approach that demands extensive memory and computational resources. We argue that resource requirements can be reduced by designing an efficient data structure to store and process subgraphs. In this paper, we propose Subgraph-To-Node (S2N) translation, a novel formulation to learn representations of subgraphs efficiently. Specifically, given a set of subgraphs in the global graph, we construct a new graph by coarsely transforming subgraphs into nodes. We theoretically and empirically show that S2N significantly reduces memory and computational costs compared to using state-of-the-art models with conventional data structures. We also suggest Coarsened S2N (CoS2N), which combines S2N with graph coarsening methods for improved results in a data-scarce setting where there are not sufficient subgraphs to cover the global graph. Our experiments on four real-world benchmarks demonstrate that fined-tuned models with S2N translation can process $183 - 711$ times more subgraph samples than state-of-the-art models at a similar or better performance level.

## 1 INTRODUCTION

Subgraph representation learning has been shown to be useful for various real-world problems (Alsentzer et al., 2020; Ju et al., 2023). Current research uses the default data structures for graph-level tasks, treating the subgraph as just a subset of the global graph. Existing studies on subgraph representation learning focus on developing graph neural networks (GNNs) specialized for subgraphs (Alsentzer et al., 2020; Wang & Zhang, 2022). However, specialized models suffer from large memory and computational requirements by performing complex operations on a large global graph. In this paper, we pose a more basic but underexplored question for subgraph representation learning prior to designing the models: *How can we effectively and efficiently store and process subgraphs as data?*

In this paper, we propose 'Subgraph-To-Node (S2N)', a novel data structure to solve subgraph-level prediction tasks efficiently. This data structure is a new graph translated from the original global graph and subgraphs, where its nodes are the original subgraphs, and its edges are the relations among the original subgraphs. Then, we can get the results of the subgraph-level tasks by performing node-level tasks from these node representations.

For example, in a knowledge graph where subgraphs are diseases, nodes are symptoms, and edges are relations between symptoms based on knowledge in the medical domain, the goal of the diagnosis task is to predict the type of a disease (Alsentzer et al., 2020). Using S2N translation, we can make a new graph of diseases, nodes of which are diseases and edges of which are relations between them (e.g., whether two diseases share symptoms).

As a benefit in return, the S2N translation enables efficient subgraph representation learning. The number of nodes in the S2N graph is decreased to the number of original subgraphs. The edges of the S2N graph are also significantly reduced, which we theoretically prove and empirically confirm in the real-world datasets. We can load large batches of subgraphs on the GPU memory and parallelize the training and inference. Since S2N translation does not interfere with model selection, even simple GNNs without complex operations can encode node representations in the translated graph.

There can be various implementations of S2N translation, and here, we create new edges as the number of shared edges across a pair of subgraphs. This method approximates the structure of the global graph into weighted edges between subgraphs; thus, part of the structural information held by the global graph is lost. We can obtain a coarse graph sufficiently informative for the task by properly normalizing edge weights. Also, we can reduce information loss by preserving the internal structure of subgraphs, which practically requires negligible computational and memory resources.

Furthermore, we address S2N's challenge when there are not sufficient samples available, specifically, representing parts of the global graph not covered by existing subgraphs. We introduce the Coarsened S2N (CoS2N), which uses graph coarsening to create 'virtual' subgraphs that summarize the global structure. The CoS2N allows message-passing between distant subgraphs with labels without compromising efficiency. We also theoretically show that CoS2N can reduce the approximation error in S2N's representations.

We conduct experiments with four real-world datasets to evaluate the performance and efficiency of S2N translation. We investigate the number of parameters, max allocated GPU memory, throughput (samples per second), and latency (seconds per forward pass) for efficiency (Dehghani et al., 2022). We demonstrate that models with S2N translation are more efficient than the existing approach without a performance drop. Specifically, while best-tuned models with S2N can process $183 - 711$ times as many samples, their performance shows $99.9 - 102.9\%$ of the state-of-the-art model.

The rest of the paper is organized as follows. First, we present a Subgraph-To-Node (S2N) translation, a novel way to generate an efficient data structure for subgraph representation learning (§3). This section includes Coarsened S2N (CoS2N), the combination with graph coarsening to tackle a data-scarce setting. Second, we theoretically show that S2N reduces the computational complexity and approximates subgraph representations from the original global graph (§4). Third, we demonstrate the efficiency of S2N compared to the state-of-the-art approaches, specifically enabling up to 711 times the throughput while maintaining the performance of at least 99.9% (§5, §6).

## 2 RELATED WORK

Our S2N translation tackles representation learning of subgraphs, and this is closely linked to graph coarsening methods. This section introduces these two fields and their connection with our study.

**Subgraph Representation Learning**  There have been various approaches to use subgraphs for expressiveness (Morris et al., 2019; Bouritsas et al., 2020), scalability (Hamilton et al., 2017; Zeng et al., 2020), augmentation (Qiu et al., 2020; You et al., 2020), modeling long-range interactions (Zhang et al., 2022; He et al., 2023), and representing meaningful clusters (Jin et al., 2018; Ying et al., 2018; Fey et al., 2020). However, only a few studies deal with learning representations of subgraphs themselves. Some works have strong assumptions on the subgraph, making it difficult to generalize (Meng et al., 2018; Kim et al., 2022). The Subgraph Neural Network (SubGNN) (Alsentzer et al., 2020) is the first approach to subgraph representation learning using topology, positions, and connectivity. The GNN with LAbeling trickS for Subgraph (GLASS) (Wang & Zhang, 2022) uses a labeling trick to distinguish nodes inside and outside the subgraph and enhance the expressive power of representations. However, both SubGNN and GLASS have model designs depending on the large global graph, so high memory and computation costs are needed. Our method allows efficient learning of subgraph representations without a complex model design. We describe the detailed architectural differences in Appendix A.1.

**Graph Coarsening**  Our S2N translation summarizes subgraphs into nodes, and in that sense, it is related to graph coarsening (or summarization) methods (Loukas & Vandergheynst, 2018; Loukas, 2019; Jin et al., 2020; Deng et al., 2020; Cai et al., 2021; Huang et al., 2021; Zhou et al., 2021; Jin et al., 2022). These methods are similar to our work that aims to handle large-scale graphs efficiently. However, the perspective of graph coarsening has not been applied to subgraph-level tasks. They focus on creating coarse graphs while preserving specific properties in a given graph (e.g., spectral similarity). In addition, the super-nodes in coarse graphs are not given to existing graph coarsening methods; thus, algorithms to decide on super-nodes are required. In S2N translation, we treat subgraphs as super-nodes and can create coarse graphs with nominal costs.

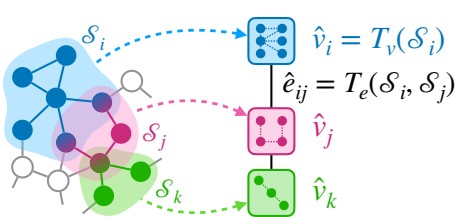

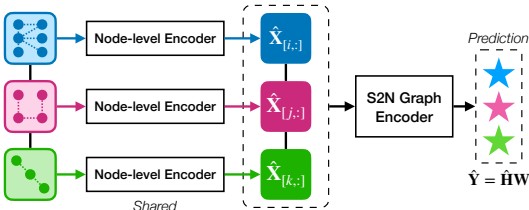

(a) The S2N translation. Subgraphs $\mathcal{S}_i$ and $\mathcal{S}_j$ are transformed into nodes $\hat{v}_i$ and $\hat{v}_j$ by $T_v$, and an edge $\hat{e}_{ij}$ between them is formed by $T_e$.

(b) Models for graphs translated by S2N. We apply a node-level encoder first (weighted sum for S2N+0 and GNN plus readout for S2N+A), then an S2N graph encoder (GNN) to their outputs for the prediction.

Figure 1: Overview of the Subgraph-To-Node (S2N) translation and models for translated graphs.

## 3 DATA STRUCTURES FOR SUBGRAPH REPRESENTATION LEARNING

We introduce three data structures for subgraph representation learning including our proposed Subgraph-To-Node (S2N) translation.

**Notations** We first summarize the notations in the subgraph representation learning, particularly in the classification task. Let $\mathcal{G} = (\mathbb{V}, \boldsymbol{A}, \boldsymbol{X})$ be a global graph where $\mathbb{V}$ is a set of nodes ($|\mathbb{V}| = N$), $\boldsymbol{A} \in \{0, 1\}^{N \times N}$ is an adjacency matrix, and $\boldsymbol{X} \in \mathbb{R}^{N \times F_0}$ is a node feature matrix. A subgraph $\mathcal{S} = (\mathbb{V}^{\text{sub}}, \boldsymbol{A}^{\text{sub}})$ is a graph formed by subsets of nodes and edges in the global graph $\mathcal{G}$. For the subgraph classification task, there is a set of $M(< N)$ subgraphs $\mathbb{S} = \{\mathcal{S}_1, \mathcal{S}_2, ..., \mathcal{S}_M\}$, and for $\mathcal{S}_i = (\mathbb{V}_i^{\text{sub}}, \boldsymbol{A}_i^{\text{sub}})$, the goal is to learn its representation $\boldsymbol{h}_i \in \mathbb{R}^F$ and the logit vector $\boldsymbol{y}_i \in \mathbb{R}^C$ where $F$ and $C$ are the numbers of hidden features and classes, respectively.

### 3.1 CONVENTIONAL DATA STRUCTURES: SEPARATED AND CONNECTED FORMS

The existing GNN-based approach employs two types of data structures when solving subgraph-level tasks. This paper refers to these two as Separated and Connected forms. The *Separated* form treats each subgraph as a separate graph, applying the GNN instance-wise for each graph. Existing studies express these separated graphs as *standalone* or *segregated* graphs and use this separated form as the main baseline. The *Connected* form represents subgraphs by applying the GNN on the whole global graph and pooling node representations. The separated form preserves only the internal structure, and the connected form retains all information in the global graph. For this reason, using the connected form requires more memory and computational resources. Since incorporating the structures in the global graph is essential in learning subgraphs, we design a new data structure that can approximate the global graph without significant costs.

### 3.2 SUBGRAPH-TO-NODE (S2N) TRANSLATION

The S2N translation reduces memory and computational costs in training and inference by constructing a new coarse graph that summarizes the original subgraph into a node. As in Figure 1a, for each subgraph $\mathcal{S}_i \in \mathbb{S}$ in the global graph $\mathcal{G}$, we create a node $\hat{v}_i = T_v(\mathcal{S}_i)$ in the translated graph $\hat{\mathcal{G}}$; for all pairs $(\mathcal{S}_i, \mathcal{S}_j)$ of subgraphs in $\mathcal{G}$, we assign an edge $\hat{e}_{ij} = T_e(\mathcal{S}_i, \mathcal{S}_j)$ between corresponding nodes in $\hat{\mathcal{G}}$. Here, $T_v$ and $T_e$ are translation functions for nodes and edges in $\hat{\mathcal{G}}$, respectively. Formally, the S2N translated graph $\hat{\mathcal{G}} = (\hat{\mathbb{V}}, \hat{\boldsymbol{A}})$ where $|\hat{\mathbb{V}}| = M$ and $\hat{\boldsymbol{A}} \in \mathbb{R}^{M \times M}$, is defined by

$$\hat{\mathbb{V}} = \{\hat{v}_i | \hat{v}_i = T_v(\mathcal{S}_i), \ \mathcal{S}_i \in \mathbb{S}\}, \ \hat{\boldsymbol{A}}_{[i,j]} = \hat{e}_{ij} = T_e(\mathcal{S}_i, \mathcal{S}_j). \quad (1)$$

We can choose any function for $T_v$ and $T_e$. For example, $T_e$ can be simple heuristics (e.g., the distance between subgraphs) or modeled with neural networks to learn the graph structure (Franceschi et al., 2019; Kim & Oh, 2021; Fatemi et al., 2021).

In this paper, we choose two versions of S2N functions with negligible translation costs: **S2N+0** and **S2N+A**. For both versions, we use the same $T_e$ to make an edge and its weight as the number of edges between two subgraphs $\mathcal{S}_i$ and $\mathcal{S}_j$, which is defined as follows:

$$T_e(\mathcal{S}_i, \mathcal{S}_j) = \sum_{v_i \in \mathbb{V}_i^{\text{sub}}} \sum_{v_j \in \mathbb{V}_j^{\text{sub}}} \boldsymbol{A}_{[v_i, v_j]}. \quad (2)$$

When using edge weights as input, if the range of the values is too wide, learning may be unstable. So, we normalize and clamp the edge weights to between 0 to 1 by selecting edges in a specific

range of standard scores ($a - b$ where $a, b$ are hyperparameters).

$$\text{normalize}(\hat{\boldsymbol{A}}) = \text{clamp}\left(\frac{(\hat{\boldsymbol{A}} - \text{mean}(\hat{\boldsymbol{A}}))/\text{std}(\hat{\boldsymbol{A}}) - a}{b - a}\right) \text{ where } \text{clamp}(x) = \max\left(0, \min\left(1, x\right)\right). \quad (3)$$

For $T_v$, we use different functions for S2N+0 and S2N+A. The difference between the two is whether it maintains the internal structures $\boldsymbol{A}_i^{\text{sub}}$ of the subgraph $\mathcal{S}_i = (\mathbb{V}_i^{\text{sub}}, \boldsymbol{A}_i^{\text{sub}})$. S2N+0 uses $T_v$ that ignores $\boldsymbol{A}_i^{\text{sub}}$ and treats the node as a set (i.e., $\mathbb{V}_i^{\text{sub}}$). In contrast, S2N+A's $T_v$ retains all information of nodes and edges in the subgraph:

$$\textbf{S2N+0: } T_v(\mathcal{S}_i) = \mathbb{V}_i^{\text{sub}}, \quad \textbf{S2N+A: } T_v(\mathcal{S}_i) = (\mathbb{V}_i^{\text{sub}}, \boldsymbol{A}_i^{\text{sub}}). \quad (4)$$

Note that their names originated from whether the adjacency matrix is a zero matrix (0) or not ($A$).

In some cases, the S2N translation provides a more intuitive description of real-world problems than a form of subgraphs. For a fitness social network (EM-User) (subgraphs: users, nodes: workouts, edges: whether multiple users complete workouts), it will be translated into a network of users connected if they complete the same workouts. This graph directly expresses the relation between users and follows the conventional approach to describe social networks where nodes are users.

### 3.3 Models for S2N Translated Graphs

We propose simple but strong models for S2N (Figure 1b): node-level encoder $\text{ENC}_{\text{node}}$ + S2N graph encoder $\text{ENC}_{\text{S2N}}$. First, $\text{ENC}_{\text{node}}$ takes $T_v(\mathcal{S})$ as an input and produces $\hat{\boldsymbol{x}}_i \in \mathbb{R}^F$, input vector for the node in the S2N graph. Then, $\text{ENC}_{\text{S2N}}$ takes $\hat{\boldsymbol{X}} = [\hat{\boldsymbol{x}}_1, ..., \hat{\boldsymbol{x}}_M]^\top \in \mathbb{R}^{M \times F}$ and $\hat{\boldsymbol{A}}$ as inputs, and produces representations $\hat{\boldsymbol{H}} = [\hat{\boldsymbol{h}}_1, ..., \hat{\boldsymbol{h}}_M]^\top \in \mathbb{R}^{M \times F}$ and logits $\hat{\boldsymbol{Y}} = [\hat{\boldsymbol{y}}_1, ..., \hat{\boldsymbol{y}}_M]^\top \in \mathbb{R}^{M \times C}$.

For $\text{ENC}_{\text{node}}$, we use different models for S2N+0 and S2N+A. Since the node in S2N+0 is a set of original nodes in $\mathcal{S}_i$, we take a set of node features in $\mathbb{V}_i$ as an input and generate a weighted sum of them. For S2N+A, we apply a GNN model to each subgraph as an individual graph, then apply a weighted sum for readout. Formally,

$$\textbf{S2N+0: } \hat{\boldsymbol{x}}_i = \sum_{v \in \mathbb{V}_i^{\text{sub}}} \omega_{vi} \cdot \boldsymbol{X}_{[v,:]}, \quad \textbf{S2N+A: } \hat{\boldsymbol{x}}_i = \sum_{v \in \mathbb{V}_i^{\text{sub}}} \omega_{vi} \cdot \text{GNN}_{\text{node}}\left(\boldsymbol{X}_{[\mathbb{V}_i^{\text{sub}},:]}, \boldsymbol{A}_i^{\text{sub}}\right)_{[v,:]}, \quad (5)$$

where $\omega_{vi}$ is a weight corresponding to the node $v$ and the subgraph $\mathcal{S}_i$. These weights can be either learnable or constants (e.g., $\omega_{*,*} = 1$ means that $\hat{\boldsymbol{x}}$ is the sum of features).

Given $\hat{\boldsymbol{A}}$ and $\hat{\boldsymbol{X}}$ of S2N+0 and S2N+A, we apply the S2N graph encoder $\text{ENC}_{\text{S2N}}$ which is another $\text{GNN}_{\text{S2N}}$ to generate the final node representations $\hat{\boldsymbol{H}}$ and logits $\hat{\boldsymbol{Y}}$ for prediction, That is,

$$\hat{\boldsymbol{H}} = \text{GNN}_{\text{S2N}}(\hat{\boldsymbol{X}}, \hat{\boldsymbol{A}}), \quad \hat{\boldsymbol{Y}} = \hat{\boldsymbol{H}}\boldsymbol{W} \text{ where } \boldsymbol{W} \in \mathbb{R}^{F \times C} \text{ is a matrix of parameters.} \quad (6)$$

We can take any GNNs that perform message-passing between nodes. This node-level message-passing on translated graphs is analogous to message-passing at the subgraph level in Sub-GNN (Alsentzer et al., 2020).

### 3.4 S2N with Graph Coarsening for a Data-Scarce Setting

By design, the S2N graph $\hat{\mathcal{G}}$ can approximate the global graph $\mathcal{G}$ covered by subgraphs, but cannot reflect parts of $\mathcal{G}$ where subgraphs do not exist. When a pair of subgraphs is distant on the global graph, they exist as unconnected nodes in S2N graphs as illustrated in Figure 2. These isolated subgraphs are likely to occur when the subgraph samples are scarce. In this case, GNNs cannot exchange supervised signals between subgraphs.

Figure 2: Overview of Subgraph-To-Node Translation with virtual subgraphs generated by graph coarsening.

To solve this problem, we apply graph coarsening methods to the global graph $\mathcal{G}$ to generate a partition of nodes in $\mathcal{G}$. That is, graph coarsening summarizes $\mathcal{G}$ by assigning one super-node to each node in $\mathcal{G}$. We construct induced subgraphs $\mathbb{S}^{\text{co}} = \{\mathcal{S}_1^{\text{co}}, \mathcal{S}_2^{\text{co}}, ..., \mathcal{S}_{M^{\text{co}}}^{\text{co}}\}$ of the global graph per a super-node. Here, we call them 'virtual subgraphs'. Using the original (labeled) subgraphs $\mathbb{S}$ as is, the virtual subgraphs are merged with $\mathbb{S}$ to form the Coarsened S2N (CoS2N) graph, formally,

$$\mathbb{S}^{\text{co}} = \text{Coarsening}(\mathcal{G}), \quad \hat{\boldsymbol{A}}_{[i,j]}^{\text{co}} = T_e(\mathcal{S}_i, \mathcal{S}_j) \text{ where } (\mathcal{S}_i, \mathcal{S}_j) \in (\mathbb{S} \cup \mathbb{S}^{\text{co}}) \times (\mathbb{S} \cup \mathbb{S}^{\text{co}}). \quad (7)$$

Training of CoS2N is done similarly to semi-supervised node classification. The virtual (unlabeled) subgraphs act as bridges to pass messages between labeled subgraphs. These allow S2N to better approximate the global graph that the existing set of subgraphs does not cover. We also show that adding virtual subgraphs to S2N can reduce the approximation error between representations of S2N and the global graph (Proposition 3).

The graph coarsening does not impair the efficiency for two reasons. First, it is performed only once before the training. Second, we can create a small CoS2N graph by tuning coarsening methods and their hyperparameters (e.g., the coarsening ratio).

## 4 THEORETICAL ANALYSIS ON S2N'S EFFICIENCY AND REPRESENTATION

This section analytically compares the efficiency and the representation quality between S2N and the original graph. We first show how much S2N reduces computational complexity. Then, we analyze the error bound of representations between S2N and the original graph when using graph convolutional networks (GCNs) (Kipf & Welling, 2017). All proofs are provided in Appendix A.3.

### 4.1 HOW MUCH DOES S2N REDUCE COMPUTATIONAL COMPLEXITY?

We first introduce more notations for this analysis. For the global graph $\mathcal{G}$ and the S2N graph $\hat{\mathcal{G}}$, the numbers of edges are $E$ and $\hat{E}$. Across a set $\mathbb{S}$ of subgraphs, the average numbers of nodes and edges are $\overline{N^{\text{sub}}}$ and $\overline{E^{\text{sub}}}$. Note that $N$ is the number of nodes in $\mathcal{G}$ and $M$ is the number of subgraphs.

In Proposition 1, we compare the time complexity of single-layer GLASS (the state-of-the-art model) (Wang & Zhang, 2022), Connected form, S2N+0, and S2N+A.

**Proposition 1.** *The time complexity of the 1-layer GLASS, Connected form, S2N+0, and S2N+A is*

| GLASS & Connected | S2N+0 | S2N+A |
|---|---|---|
| $O\left(EF + M\overline{N^{\text{sub}}}F + NF^2\right)$ | $O\left(\hat{E}F + M\overline{N^{\text{sub}}}F + MF^2\right)$ | $O\left(\hat{E}F + M\overline{E^{\text{sub}}}F + M\overline{N^{\text{sub}}}F^2\right)$ |

Considering that $N \ll E$ in real-world graphs (Chung, 2010), the significant difference between baselines and S2N is that $E$ becomes $\hat{E}$. We can know that $\hat{E}$ cannot be higher than $M^2$. The smaller $\overline{N^{\text{sub}}}$, the smaller $\hat{E}$ since it lowers the number of possible connections between nodes in subgraphs. However, it is difficult to directly compare $\hat{E}$ and $E$ without assumptions of the global graph and subgraphs. In particular, when unimportant edges of small weights are removed by normalization (Equation 3), $\hat{E}$ can be smaller than the original one.

Instead, we can gain insight from what $\hat{\boldsymbol{A}}$ takes in simple random graph models since $\hat{E}$ is the number of positive elements in $\hat{\boldsymbol{A}}$. Specifically, we employ the Configuration Model (CM) as a global graph with independent and identically distributed (i.i.d.) subgraphs. The CM graph of $N$ nodes is randomly generated from a given degree sequence $[d_1, d_2, ..., d_N]$ (Newman, 2018). We choose CM since the distribution of S2N's edge weights can be derived from the degree distribution of the global graph. See Appendix A.2 for a detailed explanation. We now demonstrate the probability that an edge weight is higher than a certain value.

**Proposition 2.** *For Configuration Model of a degree sequence $[d_1, d_2, ..., d_N]$ as $\mathcal{G}$ and i.i.d. sampled subgraphs where the average size is $\overline{N^{sub}}$, the probability that the weight $\hat{\boldsymbol{A}}_{[i,j]}$ of an edge $(i,j)$ in $\hat{\mathcal{G}}$ is bigger than $c > 0$ is $P(\hat{\boldsymbol{A}}_{[i,j]} \geq c) \leq \frac{(\overline{N^{sub}})^2 \mathbb{E}[d]}{cN}$ where $\mathbb{E}[d]$ is an average degree.*

It is well-known that degrees follow a power-law distribution in many real-world graphs (Barabási & Albert, 1999). Most nodes have a low degree; thus, the average degree $\mathbb{E}[d]$ has a small value. Proposition 2 implies that edges with small weights are more likely to appear in S2N, and the edge normalization can make the S2N graph sparse (i.e., small $\hat{E}$). We also empirically confirm that edges in S2N are fewer than those of the global graph in Figure 3.

### 4.2 HOW DOES S2N APPROXIMATE SUBGRAPH REPRESENTATIONS WHEN USING GCNS?

For this subsection, we define the mapping matrix $\boldsymbol{M} \in \{0, 1\}^{N \times M}$, where $\boldsymbol{M}_{[v,i]}$ is 1 if and only if the node $v$ belongs to the subgraph $\mathcal{S}_i$ (i.e., $\hat{\boldsymbol{A}} = \boldsymbol{M}^\top \boldsymbol{A} \boldsymbol{M}$). Degree matrices of $\mathcal{G}$ and $\hat{\mathcal{G}}$ are $\boldsymbol{D} = \text{diag}(d_1, d_2, ..., d_N)$ and $\hat{\boldsymbol{D}} = \text{diag}(\hat{d}_1, \hat{d}_2, ..., \hat{d}_M)$. Also, $\|\cdot\|$ is the Frobenius norm.

This analysis aims to analytically compare node representations $\hat{\boldsymbol{H}} \in \mathbb{R}^{M \times F}$ of the S2N graph $\hat{\mathcal{G}}$ and subgraph representations of the global graph $\mathcal{G}$. Since outputs of GNN with the global graph are original nodes' representations $\boldsymbol{H} \in \mathbb{R}^{N \times F}$, we apply the readout to pool nodes in the subgraph:

$$\text{READOUT}(\boldsymbol{H}) = \boldsymbol{R}^{\top} \boldsymbol{H} \in \mathbb{R}^{M \times F} \quad \text{where } \boldsymbol{R} \in \mathbb{R}^{N \times M} \text{ is a readout matrix.} \tag{8}$$

In this paper, we adopt a degree-dependent readout matrix $\boldsymbol{R}$ inspired by configuration-based reconstruction (Zhou et al., 2021; 2023), which is defined as follows:

$$\boldsymbol{R} = \boldsymbol{D}^{\frac{1}{2}} \boldsymbol{M} \hat{\boldsymbol{D}}^{-\frac{1}{2}} \in \mathbb{R}^{N \times M} \quad \text{i.e.,} \quad \boldsymbol{R}_{[v,i]} = (d_v / \hat{d}_i)^{\frac{1}{2}}. \tag{9}$$

We now demonstrate that the S2N's node representations $\hat{\boldsymbol{H}}$ approximate the global graph's subgraph representations $\boldsymbol{R}^{\top} \boldsymbol{H}$, particularly when the model is a single-layer GCN. The error bound between $\hat{\boldsymbol{H}}$ and $\boldsymbol{R}^{\top} \boldsymbol{H}$ is introduced in Proposition 3. We also conduct a similar analysis for a variant of Graph Isomorphism Networks (Xu et al., 2019) in Corollary 1 (Appendix A.3).

**Proposition 3.** *Using the single-layer GCN parametrized by $\boldsymbol{W}$, subgraph representations $\boldsymbol{R}^{\top} \boldsymbol{H}$ of the global graph $\mathcal{G}$ can be approximated by node representations $\hat{\boldsymbol{H}}$ of the S2N graph $\hat{\mathcal{G}}$, that is, $\hat{\boldsymbol{H}} \approx \boldsymbol{R}^{\top} \boldsymbol{H}$. The error between two representations is bounded by:*

$$\|\boldsymbol{R}^{\top} \boldsymbol{H} - \hat{\boldsymbol{H}}\| \leq M^{\frac{1}{2}} \|\boldsymbol{X} - \boldsymbol{R}\hat{\boldsymbol{X}}\| \cdot \|\boldsymbol{W}\|. \tag{10}$$

The error between representations is bounded by the error between input features $\boldsymbol{X}$ and $\boldsymbol{R}\hat{\boldsymbol{X}}$. As in Zhou et al. (2023), if we use the initial features $\hat{\boldsymbol{X}} = \boldsymbol{R}^{\top} \boldsymbol{X}$ for S2N, $\boldsymbol{R}\hat{\boldsymbol{X}}$ is $(\boldsymbol{R}\boldsymbol{R}^{\top})\boldsymbol{X}$. The matrix $\boldsymbol{R}\boldsymbol{R}^{\top} \in \mathbb{R}^{N \times N}$ has rank $M$, which is smaller than $N$, then $\boldsymbol{R}\hat{\boldsymbol{X}}$ is a low-rank approximation of $\boldsymbol{X}$. Since $\boldsymbol{R}$ is given by subgraphs, $\boldsymbol{R}\hat{\boldsymbol{X}}$ may not sufficiently approximate $\boldsymbol{X}$ for the downstream task. In particular, when there are only a few subgraph samples (i.e., very small rank $M$), the expressiveness of S2N can be weakened. This theoretical observation implies that the proposed CoS2N (§3.4) better approximates $\boldsymbol{X}$ for a data-scarce setting.

## 5 EXPERIMENTS

This section describes the experimental setup, including datasets, training, evaluation, and models.

**Datasets** We use four real-world datasets (PPI-BP, HPO-Neuro, HPO-Metab, and EM-User) and four synthetic datasets (Density, Cut-Ratio, Coreness, and Component) introduced in Alsentzer et al. (2020). The task is subgraph classification where the global graph $\mathcal{G}$ and subgraphs $\mathbb{S}$ are given in datasets. The input node features $\boldsymbol{X}$ are pre-trained embedding from Wang & Zhang (2022) for real-world datasets, and constant features and Random Walk Positional Encoding (Dwivedi et al., 2022) for synthetic datasets. Dataset statistics and descriptions are in Table 4, 5 and Appendix A.4.

**Training and Evaluation** In the original setting from Alsentzer et al. (2020), evaluation (i.e., validation and test) subgraphs cannot be seen during the training stage. Following this protocol, we create different S2N graphs for each stage using train and evaluation sets of subgraphs ($\mathbb{S}_{\text{train}}$ and $\mathbb{S}_{\text{eval}}$). For the S2N translation, we use $\mathbb{S}_{\text{train}}$ only in the training stage and use both $\mathbb{S}_{\text{train}} \cup \mathbb{S}_{\text{eval}}$ in the evaluation stage. We predict unseen nodes based on structures translated from $\mathbb{S}_{\text{train}} \cup \mathbb{S}_{\text{eval}}$ in the evaluation stage. In this respect, node classification on S2N-translated graphs is inductive.

**Models** We use two well-known GNNs for $\text{GNN}_{\text{S2N}}$: GCN (Kipf & Welling, 2017) and GC-NII (Chen et al., 2020). Note that GCNII performs well in non-homophilous graphs. For the node-level encoder $\text{GNN}_{\text{node}}$ in S2N+A, we use the same kind of GNN as $\text{GNN}_{\text{S2N}}$. See Appendix A.5 for their hyperparameters. We also test these models for Connected and Separated forms.

**Baselines** We use basic and state-of-the-art models for subgraph classification tasks as baselines: Sub2Vec (Adhikari et al., 2018), GBDT (Chen & Guestrin, 2016), SubGNN (Alsentzer et al., 2020), and GLASS (Wang & Zhang, 2022). We report the best performance among the three variants of Sub2Vec and the performance of SubGNN with pre-trained embedding by GIN. All baseline results are reprinted from Alsentzer et al. (2020) and Wang & Zhang (2022).

Table 1: Mean performance in micro F1-score over ten runs. For the top 50% of results, the higher the performance, the darker the blue color. The unpaired *t*-test result between S2N and the best is denoted by superscripts ($\sim$: no significant difference at a level of 0.01). We mark with daggers the reprinted results from Alsentzer et al. (2020) ($\dagger$) and Wang & Zhang (2022) ($\ddagger$).

| Model | Data Structure | PPI-BP | HPO-Neuro | HPO-Metab | EM-User |
|---|---|---|---|---|---|
| Sub2Vec Best$^\dagger$ | | $30.9_{\pm2.3}$ | $22.3_{\pm6.5}$ | $13.2_{\pm4.7}$ | $85.9_{\pm1.4}$ |
| GBDT$^\ddagger$ | | $44.6_{\pm0.0}$ | $51.3_{\pm0.0}$ | $40.4_{\pm0.0}$ | $69.4_{\pm0.0}$ |
| SubGNN$^\dagger$ | | $59.9_{\pm2.4}$ | $63.2_{\pm1.0}$ | $53.7_{\pm2.3}$ | $81.4_{\pm4.6}$ |
| GLASS$^\ddagger$ | | $61.9_{\pm0.7}$ | $68.5_{\pm0.5}$ | $61.4_{\pm0.5}$ | $88.8_{\pm0.6}$ |
| GCN | Separated | $61.4_{\pm2.0}$ | $67.6_{\pm1.0}$ | $60.1_{\pm2.8}$ | $84.5_{\pm4.1}$ |
| GCNII | Separated | $61.3_{\pm1.2}$ | $67.7_{\pm0.6}$ | $59.4_{\pm2.7}$ | $84.7_{\pm4.1}$ |
| GCN | Connected | $62.6_{\pm1.7}$ | $65.7_{\pm0.8}$ | $60.6_{\pm2.0}$ | $85.9_{\pm2.8}$ |
| GCNII | Connected | $63.5_{\pm2.0}$ | $66.7_{\pm0.8}$ | $61.7_{\pm2.7}$ | $85.5_{\pm4.8}$ |
| GCN | S2N+0 | $63.0^{\sim}_{\pm2.3}$ | $66.4_{\pm0.7}$ | $62.0^{\sim}_{\pm1.6}$ | $85.7_{\pm2.9}$ |
| GCNII | S2N+0 | $63.5^{\sim}_{\pm2.4}$ | $66.4_{\pm1.1}$ | $61.6^{\sim}_{\pm1.7}$ | $86.5^{\sim}_{\pm3.2}$ |
| GCN | S2N+A | $63.3^{\sim}_{\pm2.3}$ | $68.3^{\sim}_{\pm0.9}$ | $62.0^{\sim}_{\pm3.0}$ | $86.5_{\pm2.3}$ |
| GCNII | S2N+A | $63.7^{\sim}_{\pm2.3}$ | $68.4^{\sim}_{\pm1.0}$ | $63.2^{\sim}_{\pm2.7}$ | $89.0^{\sim}_{\pm1.6}$ |

**Efficiency Measurement**  We use the best hyperparameters (including batch sizes) for each model and take the mean wall-clock time over 50 epochs. Throughput and latency are all measured using training and validation sets for each stage. We count the number of all trainable parameters, including node embeddings. The maximum allocated GPU VRAM is measured by the PyTorch API. We fix the computation device as Intel(R) Xeon(R) CPU E5-2640 v4 and a single GeForce GTX 1080 Ti in measuring efficiency metrics. We describe details in Appendix A.6.

**Data-Scarce Experiments**  Experiments in a data-scarce setting are conducted on the smallest and largest graphs (PPI-BP and EM-User), and we set the number of training samples per class to 5, 10, 20, 30, and 40. To coarse the global graph, we employ the Variation Edges method (Loukas, 2019) and select the coarsening ratio that generates subgraphs smaller than average sizes. All experiments use the GCNII model, which performs well across datasets in a fully supervised setting.

## 6 RESULTS AND DISCUSSIONS

We analyze the characteristics of S2N graphs and compare our models and baselines on classification performance and efficiency. We show that S2N translation results in graph compression (§6.1), which results in a negligible decrease in classification accuracy (§6.2) but leads to significant improvements in efficiency in terms of computation and memory (§6.3). Finally, we study Coarsened S2N (CoS2N)'s performance and efficiency in a data-scare setting (§6.4).

### 6.1 ANALYSIS OF S2N-TRANSLATED GRAPHS

Figure 3 summarizes the number of nodes and edges before and after S2N translation. These statistics are from S2N graphs (S2N+0 and S2N+A) tuned for the best performance on GCN and GCNII. The translated graphs have a smaller number of nodes ($\times0.006 - \times0.27$) and edges ($\times10^{-4} - \times0.45$) than the original graphs (i.e., the connected form). We also find that they are non-homophilous, meaning many connected node pairs differ in their class. The edge homophily of S2N graphs is $0.25 \pm 0.01$ for PPI-BP, $0.20 \pm 0.03$ for HPO-Neuro[1], $0.24 \pm 0.01$ for HPO-Metab, and $0.51 \pm 0.01$ for EM-User.

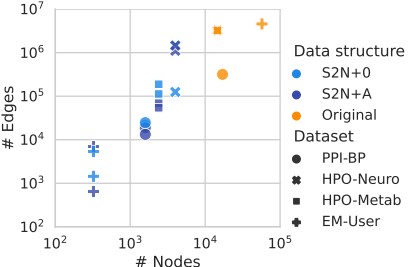

Figure 3: The number of nodes and edges of real-world graphs before and after S2N translation.

### 6.2 PERFORMANCE

**Real-World Datasets**  In Table 1, we report the mean and standard deviation of the micro F1-score over ten runs. We confirm that S2N with simple GNN models is similar to or outperforms GLASS,

---

[1]We propose multi-label edge homophily for multi-label datasets (HPO-Neuro). They generalize the existing multi-class homophily, and we discuss more in Appendix A.7.

Table 2: Mean performance in micro F1-score on synthetic datasets over 10 runs. We mark with daggers the reprinted results from Alsentzer et al. (2020) (†) and Wang & Zhang (2022) (‡).

| Model | Density | Cut-Ratio | Coreness | Component |
|---|---|---|---|---|
| Sub2Vec[‡] | $45.9_{\pm1.2}$ | $35.4_{\pm1.4}$ | $36.0_{\pm1.9}$ | $65.7_{\pm1.7}$ |
| SubGNN[†] | $91.9_{\pm1.6}$ | $62.9_{\pm3.9}$ | $65.9_{\pm9.2}$ | $95.8_{\pm9.8}$ |
| GLASS[‡] | $93.0_{\pm0.9}$ | $93.5_{\pm0.6}$ | $84.0_{\pm0.9}$ | $100.0_{\pm0.0}$ |
| GCNII / S2N+0 | $67.2_{\pm2.4}$ | $56.0_{\pm0.0}$ | $57.0_{\pm4.9}$ | $100.0_{\pm0.0}$ |
| GCNII / S2N+A | $93.2_{\pm2.6}$ | $56.0_{\pm0.0}$ | $85.7_{\pm5.8}$ | $100.0_{\pm0.0}$ |
| GCNII / S2N+0 + RWPE | $74.8_{\pm3.6}$ | $85.2_{\pm5.1}$ | $56.1_{\pm3.0}$ | $100.0_{\pm0.0}$ |
| GCNII / S2N+A + RWPE | $93.6_{\pm2.0}$ | $89.2_{\pm2.6}$ | $77.4_{\pm9.1}$ | $100.0_{\pm0.0}$ |

Table 3: The attributes that affect the subgraph properties (labels) of synthetic datasets.

| Density | Cut-Ratio | Coreness | Component |
|---|---|---|---|
| Internal structure | Border structure | Internal structure, border structure & position | Internal & external position |

the state-of-the-art model. In 16 experiments (4 datasets and 4 models), S2N models outperform GLASS in 9 cases, and SubGNN in all 16 cases. Moreover, S2N models are on par with the SOTA in 13 of 16 experiments, that is, have no significant difference from the unpaired $t$-test at the level of 0.01. The best models with S2N show 102.9% (PPI-BP), 99.9% (HPO-Neuro), 102.9% (HPO-Metab), and 100.2% (EM-User) of the performance of GLASS. We interpret that message-passing between subgraphs in S2N improves performance by capturing distant interactions that cannot occur in message-passing between nodes in the global graph. Plus, S2N+A outperforms S2N+0, that is, internal structure also contributes to subgraph representation. However, the importance of internal structures varies across datasets. Where the separated form shows relatively high performance (HPO-Neuro), the performance improvement of S2N+A over S2N+0 is high compared to other datasets.

**Synthetic Datasets** In Table 2, we summarize the performance of S2N models with GCNII and baselines on synthetic datasets. S2N+A outperforms the state-of-the-art (GLASS) on Density, Coreness, and Component datasets. S2N+0 shows the same performance as GLASS only in Component. As illustrated in Table 3, the attributes that affect the subgraph properties (i.e., labels of synthetic datasets) are known. Because S2N compresses the global graph structure, it is challenging to learn Cut-Ratio, which requires exact information about the border (or global) structure. Learning the density and coreness of subgraphs requires their internal structures. Therefore, S2N+0, which does not maintain internal structure, relatively underperforms baselines.

We can add structural encoding to the input features, particularly Random Walk Positional Encoding (RWPE) (Dwivedi et al., 2022) to address this issue. The efficiency of S2N is maintained since the RWPE is computed once before training and only requires the memory complexity of $O(N)$. As shown in the last two rows of Table 2, RWPE allows S2N to significantly improve the performance of Density and Cut-Ratio , but not of Coreness. We interpret that RWPE for subgraphs can encode internal and border structures well but cannot encode border positions. We leave the development of structural encoding for S2N as future work.

## 6.3 EFFICIENCY

In Figure 4, we show throughput (subgraphs / second), latency (seconds / forward pass), the number of parameters, and the maximum allocated GPU VRAM of two models with three data structures and state-of-the-art baselines. We cannot experiment on PPI-BP with SubGNN since it takes more than 48 hours in pre-computation. We make the following five observations from these results.

**S2N models show significantly high throughput (Figure 4a).** The best S2N models can process $\times183 - \times711$ more samples than the state-of-the-art model (GLASS) for the same training time. At the evaluation stage, they show $7 - 56$ times higher throughput than GLASS. This difference is not as large as the training stage, but S2N is still significantly more efficient than GLASS. In addition, S2N shows higher training throughput than connected and separated forms.

**S2N models even with full batch show lower latency than others with small batch size (Figure 4b).** Comparing the best S2N model and GLASS, the training latency is $\times0.05 - \times0.17$ and evaluation latency is $\times0.16 - \times0.43$. Note that measuring latency ignores the parallelism from large

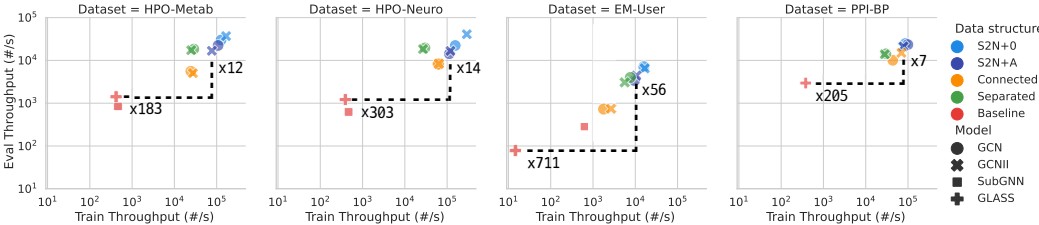

(a) The throughput (the number of subgraphs / second) at training and evaluation stages. *The higher, the better.*

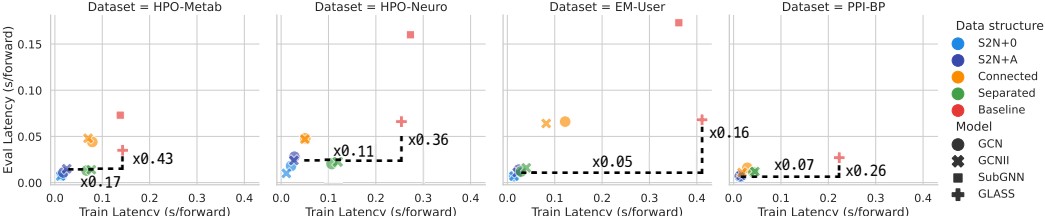

(b) The latency (seconds / forward pass) at training and evaluation stages. *The lower, the better.*

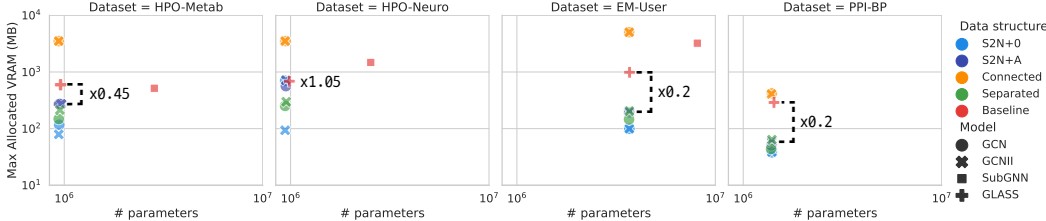

(c) The number of parameters and maximum allocated GPU VRAM. *The lower, the better.*

Figure 4: Efficiency of S2N models and baselines on real-world datasets. The ratio of each metric of the best S2N model and the state-of-the-art model is notated in the figure (dashed lines).

batch sizes. S2N's superiority over other data structures can be underestimated in latency rather than throughput because it requires full batch computation. Note that existing models should use small batch sizes by intensive memory requirements (SubGNN) or model design (GLASS).

**S2N models require less memory even with a similar level of parameters (Figure 4c).** For a given dataset, the number of parameters of each model does not vary much, but the GPU VRAM in the actual runtime varies by a large margin. The best models with S2N need less memory ($\times 0.2 - \times 0.45$) than GLASS except for HPO-Neuro. For HPO-Neuro, which has a large number of subgraphs, requires the same level of memory ($\times 1.05$). In particular, since S2N does not employ a large global graph, S2N works with only $\times 0.13$ memory on average compared to the connected form.

**S2N+A does not show a significant difference from S2N+0 in efficiency.** Recall that S2N+A differs from S2N+0 by using the internal edges of subgraphs. However, the number of internal edges is negligible compared to the original global edges, as in Table 4. Consequently, the added internal edges require only a small amount of additional computation and memory, allowing S2N+A to perform training and inference efficiently.

**Overall, S2N models outperform baselines in all computational and memory efficiency metrics.** Models with S2N process many samples faster (i.e., higher throughput and lower latency), and require less GPU memory than other data structures and state-of-the-art models. The separated form, which does not use a global graph, shows a similar level of efficiency as S2N in some experiments but loses performance by ignoring the global structure completely.

### 6.4 Performance and Efficiency of Coarsened S2N in a Data-Scarce Setting

This section reports the performance and efficiency of data structures: connected, separated forms, and Coarsened S2N (CoS2N). Figure 5 summarizes the performance, training and evaluation

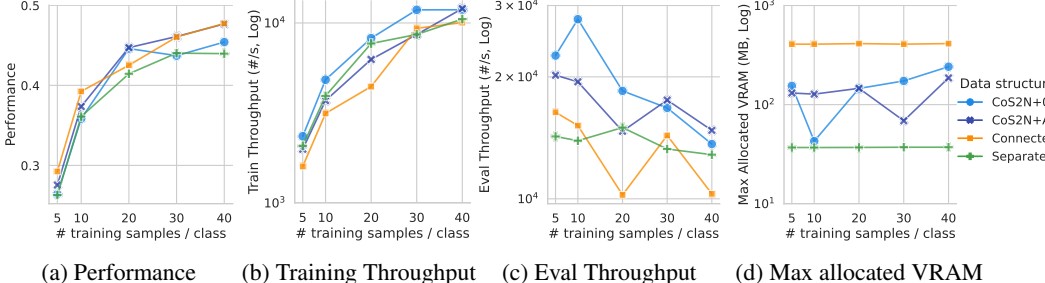

(a) Performance     (b) Training Throughput     (c) Eval Throughput     (d) Max allocated VRAM

Figure 5: Performance and efficiency on PPI-BP of CoS2N, connected, and separated forms by the number of training samples in a data-scarce setting.

throughput, and max allocated VRAM by the number of training samples on PPI-BP. In Appendix A.10, we discuss the results of the other dataset and ablation study on the coarsening ratio.

**Subgraphs created by coarsening contribute to performance improvements of S2N (Figure 5a).** CoS2N+A outperforms the separated form in all conditions. Note that the separated form in a data-scarce setting is identical to S2N+A without coarsening, which is the weakest baseline that investigates the effectiveness of coarsening. This implies that virtual subgraphs created by coarsening help to pass messages between nodes in S2N, resulting in better representations. Moreover, CoS2N+A shows similar performance to the connected form and even outperforms when the number of samples is 20. We confirm that CoS2N approximates representations of the global graph well, even though the virtual subgraphs created through coarsening do not follow the distribution of real subgraphs.

**CoS2N has higher throughput (Figures 5b, 5c) and uses less memory (Figure 5d) than using the global graph.** Although the virtual subgraphs by coarsening are added, both CoS2N methods show higher throughput than using the global graph (i.e., the connected form). CoS2N+0 even shows higher throughput than the separated form in all stages. CoS2N+A shows higher throughput than the separated form in the evaluation stage, where there are a larger number of subgraphs to be processed. The training throughput increases as more training samples are used since the full batch parallelization of GPUs can efficiently process additional samples.

Like computational requirements, CoS2N uses less memory than the connected form. This is because graph coarsening can create a number of subgraphs smaller than the size of the global graph. For CoS2N and the connected form, the memory consumption is constant or fluctuates with respect to the training set size. The memory bottleneck of the connected form and CoS2N is the largest component of each dataset: the global graph and coarsened nodes. Adding training samples does not substantially affect memory demand.

## 7    CONCLUSION

Subgraph-To-Node (S2N) translation is a novel, efficient way to learn representations of subgraphs. S2N takes the original subgraphs, and creates a new graph where the nodes are the subgraphs and the edges are the relations between the subgraphs, thereby performing subgraph-level tasks as node-level tasks. We empirically and theoretically show that S2N translation significantly reduces memory and computation costs without performance degradation. Specifically, the best-performing models with S2N on real-world datasets show $\times 183 - \times 711$ of throughput and achieve at least 99.9% of the state-of-the-art models for classification performance. One future direction is to sample subgraphs to construct S2N graphs when there are too many subgraphs. We expect sampling subgraphs to prune less informative nodes and edges in S2N and further enhance efficiency.

## 8 REPRODUCIBILITY STATEMENT

To reproduce the results, we open our code public. Our code is available at `supplementary_materials`. The GitHub link to the code will be visible after the acceptance. Datasets, including the downloadable link from Alsentzer et al. (2020), are described in Appendix A.4. All proofs of theoretical claims are provided in Appendix A.3.

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

# A APPENDIX

## A.1 DETAILED DESCRIPTIONS OF ARCHITECTURAL DIFFERENCES

### A.1.1 COMPARISON WITH SUBGNN AND GLASS

SubGNN (Alsentzer et al., 2020), GLASS (Wang & Zhang, 2022), and S2N improve different parts of the machine learning pipeline to solve subgraph-level tasks. SubGNN designs a whole model, GLASS augments input data through a labeling trick, and S2N uses a new data structure.

SubGNN performs message-passing between subgraphs (or patches). Through this, the properties of internal and border structures for three channels (position, neighborhood, and structure) are learned independently. To learn a total of 6 ($2 \times 3$) properties, SubGNN designs patch samplers, patch representation, and similarity (weights of messages) for each property in an ad hoc manner. To learn internal positions, for example, SubGNN patches nodes inside the subgraph uses its representation as a message, and uses distance-based similarity as weights. By the complex model design, SubGNN requires a lot of computational resources for data pre-processing, model training, and inference.

GLASS uses plain GNNs but labels input nodes as to whether they belong to the subgraph (the label of one) or not (the label of zero). Separate node-level message-passing is performed for each label to distinguish the internal and border structures of the subgraph. GLASS's labeling trick is effective, but hard to handle multiple labels from multiple subgraphs in a batch. Although the authors of GLASS propose a max-zero-one trick to address this issue, small batches are still recommended. In addition, using a large global graph requires significant computational and memory resources.

In comparison, our proposed S2N uses the new data structure that stores and processes subgraphs efficiently. By compressing the global graph, computational and memory resource requirements are reduced. There are no restrictions on batch learning so we can train S2N graphs in the full batch.

### A.1.2 COMPARISON WITH DIFFPOOL

DiffPool (Ying et al., 2018) learns the hierarchy of a graph to obtain graph-level representations. DiffPool softly assigns each node to a cluster during training by optimizing the downstream task loss. To stabilize the soft clustering assignment, the authors of DiffPool employ link prediction loss and entropy regularization loss. The problem is that the assignment matrix must be maintained in GPU memory, which requires quadratic memory complexity regarding the number of nodes. In other words, we cannot apply DiffPool to large graphs such as global graphs in our use cases.

We aim to perform subgraph representation learning efficiently by compressing data and reducing GPU load. Memory-intensive graph coarsening, such as DiffPool's soft clustering assignment, should not be used to keep CoS2N efficient. Instead, we can secure the efficiency of CoS2N by performing graph coarsening before training the model, relying only on the structure of the global graph.

### A.1.3 COMPARISON WITH JT-VAE

Junction Tree Variational Autoencoder (JT-VAE) (Jin et al., 2018) decomposes a molecular graph into a junction tree, where a node corresponds to the motif (particularly a ring of atoms), and edges link the nodes that share the nodes. This method is a graph generation model to learn the ring substructure well in chemical tasks but has not been used in subgraph-level tasks. Due to the nature of the Junction Tree algorithm, only the ring (or cycle) structure of input graphs is used as subgraphs, and the output is restricted to trees, which leads to limited usage.

Our proposed S2N's primary contribution is to explore the fundamental question of subgraph representation learning and propose a novel perspective. In addition, S2N can be generally applied to graphs and subgraphs of any structure.

## A.2 JUSTIFICATION FOR THE CHOICE OF THE RANDOM GRAPH MODEL

The complexity of S2N strongly depends on the distribution of translated edge weights. Thus, we need a random graph model that can analytically calculate the distribution of edge weights (i.e.,

the number of shared edges in two subgraphs). When using the configuration model (CM), the distribution of S2N's edge weights can be derived from the degree distribution of the global graph. This is possible because CM calculates the probability of edge existence through the degrees of a pair of nodes. Note that CM is frequently used in analytically calculating numerous network measures (Barabási, 2013).

We also emphasize that CM only requires a degree sequence or a distribution. That means CM can also generate graphs generated by other random graph models. For example, when the degree distribution is Poisson distribution, CM generates graphs close to the Erdős–Rényi model. CM can also generate degree distributions with other distributions, for example, power-law distributions. See Newman (2018) for more details.

### A.3 PROOF OF THEORETICAL ANALYSIS

**Proposition 1.** *The time complexity of the 1-layer GLASS, Connected form, S2N+0, and S2N+A is*

| GLASS & Connected | S2N+0 | S2N+A |
|---|---|---|
| $O\left(EF + M\overline{N^{\text{sub}}}F + NF^2\right)$ | $O\left(\hat{E}F + M\overline{N^{\text{sub}}}F + MF^2\right)$ | $O\left(\hat{E}F + M\overline{E^{\text{sub}}}F + M\overline{N^{\text{sub}}}F^2\right)$ |

*Proof.* Let $\mathcal{G} = (\mathbb{V}, \boldsymbol{A}, \boldsymbol{X})$ be a global graph where $\mathbb{V}$ is a set of nodes ($|\mathbb{V}| = N$), $\boldsymbol{A} \in \{0, 1\}^{N \times N}$ is an adjacency matrix, and $\boldsymbol{X} \in \mathbb{R}^{N \times F_0}$ is a node feature matrix. A subgraph $\mathcal{S} = (\mathbb{V}^{\text{sub}}, \boldsymbol{A}^{\text{sub}})$ is a graph formed by subsets of nodes and edges in the global graph $\mathcal{G}$. For the subgraph classification task, there is a set of $M$ subgraphs $\mathbb{S} = \{\mathcal{S}_1, \mathcal{S}_2, ..., \mathcal{S}_M\}$, and for $\mathcal{S}_i = (\mathbb{V}_i^{\text{sub}}, \boldsymbol{A}_i^{\text{sub}})$, the goal is to learn subgraph representations $\hat{\boldsymbol{H}} \in \mathbb{R}^{M \times F}$.

Baselines and S2N models are computed by following steps:

- GLASS & Connected: $\hat{\boldsymbol{H}} = \boldsymbol{R}^\top \text{GNN}(\boldsymbol{X}, \boldsymbol{A})$ where $\boldsymbol{R} \in \mathbb{R}^{N \times M}$ is a readout matrix.

- S2N+0: $\hat{\boldsymbol{H}} = \text{GNN}_{\text{S2N}}(\hat{\boldsymbol{X}}, \hat{\boldsymbol{A}})$ where $\hat{\boldsymbol{X}}_{[i]} = \sum_{v \in \mathbb{V}_i^{\text{sub}}} \omega_{vi} \cdot \boldsymbol{X}_{[v,:]}$.

- S2N+A: $\hat{\boldsymbol{H}} = \text{GNN}_{\text{S2N}}(\hat{\boldsymbol{X}}, \hat{\boldsymbol{A}})$ where $\hat{\boldsymbol{X}}_{[i]} = \sum_{v \in \mathbb{V}_i^{\text{sub}}} \omega_{vi} \cdot \text{GNN}_{\text{node}}(\boldsymbol{X}_{[\mathbb{V}_i^{\text{sub}},:]}, \boldsymbol{A}_i^{\text{sub}})_{[v,:]}$.

Graph neural networks (GNNs) that use the message-passing mechanism to learn subgraph representations can be decomposed into feature transformation (FT), feature propagation (FP), and subgraph-level readout (SR). Feature transformation requires $O(\text{the number of nodes} \times F^2)$ computations and feature propagation by sparse implementation requires $O(\text{the number of edges} \times F)$ computations. Plus, for the readout of representations or input features, we need the computations of $O(\text{the total number of nodes in subgraphs} \times F)$.

- GLASS & Connected: $O(EF)$ from FP, $O(M\overline{N^{\text{sub}}}F)$ from SR, and $O(NF^2)$ from FT.

- GLASS: $O(M\overline{N^{\text{sub}}})$ from the node labeling trick (Wang & Zhang, 2022).

- S2N+0: $O(\hat{E}F)$ from FP, $O(M\overline{N^{\text{sub}}}F)$ from SR, and $O(MF^2)$ from FT.

- S2N+A: $O(\hat{E}F)$ and $O(M\overline{E^{\text{sub}}}F)$ from FP in $\text{GNN}_{\text{S2N}}$ and $\text{GNN}_{\text{node}}$, $O(M\overline{N^{\text{sub}}}F)$ from SR, and $O(MF^2)$ and $O(M\overline{N^{\text{sub}}}F^2)$ from FT in $\text{GNN}_{\text{S2N}}$ and $\text{GNN}_{\text{node}}$.

By adding up all the terms, we can get the final result. □

**Proposition 2.** *For Configuration Model of a degree sequence $[d_1, d_2, ..., d_N]$ as $\mathcal{G}$ and i.i.d. sampled subgraphs where the average size is $\overline{N^{sub}}$, the probability that the weight $\hat{\boldsymbol{A}}_{[i,j]}$ of an edge $(i, j)$ in $\hat{\mathcal{G}}$ is bigger than $c > 0$ is $P(\hat{\boldsymbol{A}}_{[i,j]} \geq c) \leq \frac{(\overline{N^{sub}})^2 \mathbb{E}[d]}{cN}$ where $\mathbb{E}[d]$ is an average degree.*

*Proof.* We first note that the probability of edge $(u, v)$ in the Configuration Model for large $E$ is $\frac{d_u d_v}{2E}$ and $E = \frac{1}{2}\sum_k d_k = \frac{1}{2}N\mathbb{E}[d]$ (Newman, 2018).

$$P(\hat{A}_{[i,j]} \geq c) \leq \mathbb{E}[\hat{A}_{[i,j]}]/c \quad (\because \text{Markov's inequality}) \tag{11}$$

$$= \mathbb{E}[\textstyle\sum_{u \in \mathbb{V}_i^{\text{sub}}} \sum_{v \in \mathbb{V}_j^{\text{sub}}} A_{[u,v]}]/c \tag{12}$$

$$= \mathbb{E}[\textstyle\sum_{u \in \mathbb{V}_i^{\text{sub}}} \sum_{v \in \mathbb{V}_j^{\text{sub}}} \frac{d_u d_v}{2E}]/c \tag{13}$$

$$= \mathbb{E}_{(i,j) \in \mathbb{S} \times \mathbb{S}}[\textstyle\sum_{u \in \mathbb{V}_i^{\text{sub}}} \sum_{v \in \mathbb{V}_j^{\text{sub}}} \mathbb{E}[d]^2]/(2cE) \tag{14}$$

$$= \frac{(\overline{N^{\text{sub}}}\mathbb{E}[d])^2}{2cE} \tag{15}$$

$$= \frac{(\overline{N^{\text{sub}}})^2\mathbb{E}[d]}{cN} \tag{16}$$

$\square$

To prove Proposition 3, we first introduce Lemma 1.

**Lemma 1.**

$$R^\top D^{-\frac{1}{2}} A D^{-\frac{1}{2}} R = \hat{D}^{-\frac{1}{2}} \hat{A} \hat{D}^{-\frac{1}{2}} \tag{17}$$

*Proof.*

$$R^\top D^{-\frac{1}{2}} A D^{-\frac{1}{2}} R \tag{18}$$

$$= (D^{\frac{1}{2}} M \hat{D}^{-\frac{1}{2}})^\top D^{-\frac{1}{2}} A D^{-\frac{1}{2}} D^{\frac{1}{2}} M \hat{D}^{-\frac{1}{2}} \tag{19}$$

$$= \hat{D}^{-\frac{1}{2}} M^\top D^{\frac{1}{2}} D^{-\frac{1}{2}} A D^{-\frac{1}{2}} D^{\frac{1}{2}} M \hat{D}^{-\frac{1}{2}} \tag{20}$$

$$= \hat{D}^{-\frac{1}{2}} M^\top A M \hat{D}^{-\frac{1}{2}} \tag{21}$$

$$= \hat{D}^{-\frac{1}{2}} \hat{A} \hat{D}^{-\frac{1}{2}}. \tag{22}$$

$\square$

**Proposition 3.** *Using the single-layer GCN parametrized by $W$, subgraph representations $R^\top H$ of the global graph $\mathcal{G}$ can be approximated by node representations $\hat{H}$ of the S2N graph $\hat{\mathcal{G}}$, that is, $\hat{H} \approx R^\top H$. The error between two representations is bounded by:*

$$\|R^\top H - \hat{H}\| \leq M^{\frac{1}{2}}\|X - R\hat{X}\| \cdot \|W\|. \tag{23}$$

*Proof.*

$$\|R^\top H - \hat{H}\| \tag{24}$$

$$= \|R^\top D^{-\frac{1}{2}} A D^{-\frac{1}{2}} X W - \hat{D}^{-\frac{1}{2}} \hat{A} \hat{D}^{-\frac{1}{2}} \hat{X} W\| \tag{25}$$

$$= \|R^\top D^{-\frac{1}{2}} A D^{-\frac{1}{2}} X W - R^\top D^{-\frac{1}{2}} A D^{-\frac{1}{2}} R\hat{X} W\| \quad (\because \text{Lemma 1}) \tag{26}$$

$$= \|R^\top (D^{-\frac{1}{2}} A D^{-\frac{1}{2}})(X - R\hat{X})W\| \tag{27}$$

$$\leq \|R^\top\|\|D^{-\frac{1}{2}} A D^{-\frac{1}{2}}\|\|X - R\hat{X}\|\|W\| \tag{28}$$

$$\leq M^{\frac{1}{2}}\|X - R\hat{X}\| \cdot \|W\|. \tag{29}$$

$\square$

Although Proposition 3 is analyzed using GCN models only, it is not limited to GCN in its applicability. Intuitively, when sufficient subgraph samples are not available, message-passing in any GNNs fails in the global graph not covered by existing subgraphs. Moreover, we can obtain theoretical results similar to Proposition 3 for other GNNs. However, we might not get the approximation bound analytically depending on GNN architectures. For Graph Isomorphism Network (GIN) (Xu et al., 2019) as an example, the non-linearity in multi-layer perceptron (MLP) makes it hard to analytically compare the GIN outputs of S2N and the original graph. Instead, we introduce an approximation

Table 4: Statistics of real-world datasets in original forms (before S2N translation).

|  | PPI-BP | HPO-Neuro | HPO-Metab | EM-User |
|---|---|---|---|---|
| # nodes in $\mathcal{G}$ | 17,080 | 14,587 | 14,587 | 57,333 |
| # edges in $\mathcal{G}$ | 316,951 | 3,238,174 | 3,238,174 | 4,573,417 |
| # internal edges in subgraphs | 9,627 | 217,555 | 390,450 | 86,648 |
| # subgraphs | 1,591 | 4,000 | 2,400 | 324 |
| Density of $\mathcal{G}$ | 0.0022 | 0.0304 | 0.0304 | 0.0028 |
| Average density of subgraphs | $0.216_{\pm 0.188}$ | $0.767_{\pm 0.141}$ | $0.757_{\pm 0.149}$ | $0.010_{\pm 0.006}$ |
| Average # nodes / subgraph | $10.2_{\pm 10.5}$ | $14.8_{\pm 6.5}$ | $14.4_{\pm 6.2}$ | $155.4_{\pm 100.2}$ |
| Average # components / subgraph | $7.0_{\pm 5.5}$ | $1.5_{\pm 0.7}$ | $1.6_{\pm 0.7}$ | $52.1_{\pm 15.3}$ |
| # classes | 6 | 10 | 6 | 2 |
| Single- or multi-label | Single-label | Multi-label | Single-label | Single-label |
| Train/Valid/Test splits | 80/10/10 | 80/10/10 | 80/10/10 | 70/15/15 |

error bound on 'GIN Sum-1-Layer', a less powerful variant of GINs that replaces MLP with single-layer perceptron (SLP).

$$\text{GIN: } \boldsymbol{H} = \text{MLP}\left((\mathbf{A} + (1+\epsilon)\cdot \mathbf{I})\cdot \boldsymbol{X}\right), \tag{30}$$

$$\text{GIN Sum-1-Layer: } \boldsymbol{H} = \text{SLP}\left((\mathbf{A} + (1+\epsilon)\cdot \mathbf{I})\cdot \boldsymbol{X}\right). \tag{31}$$

The error bound between S2N's node representations and the global graph's subgraph representations is demonstrated in Proposition 4. Here, we use the sum-readout $\text{READOUT}(\boldsymbol{H}) = \boldsymbol{M}^\top \boldsymbol{H}$ to get subgraph representations.

**Proposition 4.** *Using the single-layer GIN Sum-1-Layer parametrized by $\boldsymbol{W}$, subgraph representations $\boldsymbol{M}^\top \boldsymbol{H}$ of the global graph $\mathcal{G}$ can be approximated by node representations $\hat{\boldsymbol{H}}$ of the S2N graph $\hat{\mathcal{G}}$, that is, $\hat{\boldsymbol{H}} \approx \boldsymbol{M}^\top \boldsymbol{H}$. The error between two representations is bounded by:*

$$\|\boldsymbol{M}^\top \boldsymbol{H} - \hat{\boldsymbol{H}}\| \leq \left((M\overline{N^{sub}}E)^{\frac{1}{2}}\|\boldsymbol{X} - \boldsymbol{M}\hat{\boldsymbol{X}}\| + (1+\epsilon)\|\boldsymbol{M}^\top \boldsymbol{X} - \hat{\boldsymbol{X}}\|\right)\cdot \|\boldsymbol{W}\|. \tag{32}$$

*Proof.*

$$\|\boldsymbol{M}^\top \boldsymbol{H} - \hat{\boldsymbol{H}}\| \tag{33}$$

$$= \|\boldsymbol{M}^\top (\boldsymbol{A} + (1+\epsilon)\boldsymbol{I}_N)\boldsymbol{X}\boldsymbol{W} - (\hat{\boldsymbol{A}} + (1+\epsilon)\boldsymbol{I}_M)\hat{\boldsymbol{X}}\boldsymbol{W}\| \tag{34}$$

$$= \|\boldsymbol{M}^\top (\boldsymbol{A} + (1+\epsilon)\boldsymbol{I}_N)\boldsymbol{X}\boldsymbol{W} - (\boldsymbol{M}^\top \boldsymbol{A}\boldsymbol{M} + (1+\epsilon)\boldsymbol{I}_M)\hat{\boldsymbol{X}}\boldsymbol{W}\| \tag{35}$$

$$= \|\boldsymbol{M}^\top \boldsymbol{A}(\boldsymbol{X} - \boldsymbol{M}\hat{\boldsymbol{X}})\boldsymbol{X}\boldsymbol{W} + (1+\epsilon)(\boldsymbol{M}^\top \boldsymbol{X} - \hat{\boldsymbol{X}})\boldsymbol{W}\| \tag{36}$$

$$\leq \|\boldsymbol{M}^\top\|\|\boldsymbol{A}\|\|\boldsymbol{X} - \boldsymbol{M}\hat{\boldsymbol{X}}\|\|\boldsymbol{W}\| + (1+\epsilon)\|\boldsymbol{M}^\top \boldsymbol{X} - \hat{\boldsymbol{X}}\|\|\boldsymbol{W}\| \tag{37}$$

$$\leq \left((M\overline{N^{sub}}E)^{\frac{1}{2}}\|\boldsymbol{X} - \boldsymbol{M}\hat{\boldsymbol{X}}\| + (1+\epsilon)\|\boldsymbol{M}^\top \boldsymbol{X} - \hat{\boldsymbol{X}}\|\right)\cdot \|\boldsymbol{W}\|, \tag{38}$$

where $\boldsymbol{I}_N$ is an identity matrix of size $N$. $\square$

If we set the initial features of S2N as a sum of the original features (i.e., $\boldsymbol{X} = \boldsymbol{M}^\top \boldsymbol{X}$), Corollary 1 then follows from Proposition 4.

**Corollary 1.** *Using the single-layer GIN Sum-1-Layer parametrized by $\boldsymbol{W}$, subgraph representations $\boldsymbol{M}^\top \boldsymbol{H}$ of the global graph $\mathcal{G}$ can be approximated by node representations $\hat{\boldsymbol{H}}$ of the S2N graph $\hat{\mathcal{G}}$, that is, $\hat{\boldsymbol{H}} \approx \boldsymbol{M}^\top \boldsymbol{H}$. If the initial feature matrix of S2N is $\hat{\boldsymbol{X}} = \boldsymbol{M}^\top \boldsymbol{X}$, the error between two representations is bounded by:*

$$\|\boldsymbol{M}^\top \boldsymbol{H} - \hat{\boldsymbol{H}}\| \leq (M\overline{N^{sub}}E)^{\frac{1}{2}}\|\boldsymbol{X} - \boldsymbol{M}\hat{\boldsymbol{X}}\|\cdot \|\boldsymbol{W}\|. \tag{39}$$

### A.4 DATASETS

All real-world subgraph datasets (PPI-BP, HPO-Neuro, HPO-Metab, and EM-User) and synthetic subgraph datasets (Density, Cut-Ratio, Coreness, and Component) are proposed in Alsentzer et al. (2020).

Table 5: Statistics of synthetic datasets in original forms (before S2N translation).

|  | Density | Cut-Ratio | Coreness | Component |
|---|---|---|---|---|
| # nodes in $\mathcal{G}$ | 5,000 | 5,000 | 5,000 | 19,555 |
| # edges in $\mathcal{G}$ | 29,521 | 83,969 | 118,785 | 43,701 |
| # subgraphs | 250 | 250 | 221 | 250 |
| Density of $\mathcal{G}$ | 0.0024 | 0.0067 | 0.0095 | 0.0002 |
| Average density of subgraphs | $0.232_{\pm 0.146}$ | $0.945_{\pm 0.028}$ | $0.219_{\pm 0.062}$ | $0.150_{\pm 0.161}$ |
| Average # nodes / subgraph | $20.0_{\pm 0.0}$ | $20.0_{\pm 0.0}$ | $20.0_{\pm 0.0}$ | $74.2_{\pm 52.8}$ |
| Average # components / subgraph | $3.8_{\pm 3.7}$ | $1.0_{\pm 0.0}$ | $1.0_{\pm 0.0}$ | $4.9_{\pm 3.5}$ |
| # classes | 3 | 3 | 3 | 2 |
| Single- or multi-label | Single-label | Single-label | Single-label | Single-label |
| Train/Valid/Test splits | 80/10/10 | 80/10/10 | 80/10/10 | 80/10/10 |

They can be downloaded from the author's GitHub repository[2]. Pre-trained embeddings can be downloaded from the GitHub repository[3] of Wang & Zhang (2022). We describe their nodes, edges, subgraphs, tasks, and references in the following paragraphs. Note that the number of edges in the real-world datasets compared to datasets referred to as large-scale (Lim et al., 2021) is at a similar level; thus, similar scalability is required to model real-world graphs using GNNs.

**PPI-BP** The global graph of PPI-BP (Zitnik et al., 2018; Subramanian et al., 2005; Consortium, 2019; Ashburner et al., 2000) is a human protein-protein interaction (PPI) network; nodes are proteins, and edges are whether there is a physical interaction between proteins. Subgraphs are sets of proteins in the same biological process (e.g., alcohol bio-synthetic process). The task is to classify processes into six categories.

**HPO-Neuro and HPO-Metab** These two HPO (Human Phenotype Ontology) datasets (Hartley et al., 2020; Köhler et al., 2019; Mordaunt et al., 2020) are knowledge graphs of phenotypes (i.e., symptoms) of rare neurological and metabolic diseases. Each subgraph is a collection of symptoms associated with a monogenic disorder. The task is to diagnose the rare disease: classifying the disease type among subcategories (ten for HPO-Neuro and six for HPO-Metab).

**EM-User** EM-User (Users in EndoMondo) dataset is a social fitness network from Endomondo (Ni et al., 2019). Here, subgraphs are users, nodes are workouts, and edges exist between workouts completed by multiple users. Each subgraph represents the workout history of a user. The task is to profile a user's gender.

**Density, Cut-Ratio, Coreness, and Component** For these synthetic datasets, the task is to predict the properties of subgraphs: density, cut ratio, average core number, and the number of components, respectively. Refer to (Alsentzer et al., 2020) for details to generate the synthetic graphs. We use a vector of 64 dimensions initialized to 1 or its L1-normalized vector as input node embedding. When using RWPE, we allocate 1/2 or 1/4 of the total embedding dimension.

## A.5 Models

This section describes the hyperparameter details and the tuning method. All models are implemented with PyTorch (Paszke et al., 2019), PyTorch Geometric (Fey & Lenssen, 2019), and PyTorch Lightning (Falcon & The PyTorch Lightning team, 2019).

We tune hyperparameters using TPE (Tree-structured Parzen Estimator) algorithm in Optuna (Akiba et al., 2019) by 400 trials: learning rate ($5 \times 10^{-4} - 10^{-2}$), weight decay ($10^{-9} - 10^{-6}$), the number of layers in GNN (1 − 2), dropout of channels and edges ($\{0.0, 0.1, ..., 0.5\}$), gradient clipping ($\{0.0, 0.1, ..., 0.5\}$), the readout matrix ($\omega_{vi} = \boldsymbol{R}_{[v,i]}$ in Equation 9 or $\omega_{vi} = \boldsymbol{M}_{[v,i]}$), and whether to use batch normalization (Ioffe & Szegedy, 2015) and skip-connection (He et al., 2016). Hyperparameters specialized on GCNII are also tuned: $\alpha$ ($\{0.1, 0.2, ..., 0.9\}$), $\theta$ ($\{0.1, 0.2, ..., 2.0\}$), weight

---

[2] https://github.com/mims-harvard/SubGNN
[3] https://github.com/Xi-yuanWang/GLASS

sharing (True or False). For S2N translation, we tune edge normalization range ($a$ and $b = a + \Delta$ in Equation 3, $a \in \{1.0, 1.25, ..., 4.0\}$, $\Delta \in \{0.5, 1.0, 1.5, 2.0\}$).

All hyperparameters are reported in the code.

## A.6 EFFICIENCY MEASUREMENT

We compute throughput (subgraphs per second) and latency (seconds per forward pass) by following equations. In addition, we use `torch.cuda.max_memory_allocated` to measure the maximum allocated GPU VRAM[4].

$$\text{Training throughput} = \frac{\text{\# of training subgraphs}}{\text{training wall-clock time (seconds) / \# of epochs}}, \quad (40)$$

$$\text{Evaluation throughput} = \frac{\text{\# of validation subgraphs}}{\text{validation wall-clock time (seconds) / \# of epochs}}, \quad (41)$$

$$\text{Training latency} = \frac{\text{training wall-clock time (seconds)}}{\text{\# of training batches}}, \quad (42)$$

$$\text{Evaluation latency} = \frac{\text{validation wall-clock time (seconds)}}{\text{\# of validation batches}}. \quad (43)$$

## A.7 GENERALIZATION OF HOMOPHILY TO MULTI-LABEL CLASSIFICATION

Node (Pei et al., 2020) and edge homophily (Zhu et al., 2020) are defined by,

$$h^{\text{edge}} = \frac{|\{(u,v)|(u,v) \in \mathbb{A} \wedge y_u = y_v\}|}{|\mathbb{A}|}, \ h^{\text{node}} = \frac{1}{|\mathbb{V}|} \sum_{v \in \mathbb{V}} \frac{|\{(u,v)|u \in \mathcal{N}(v) \wedge y_u = y_v\}|}{|\mathcal{N}(v)|}, \quad (44)$$

where $y_v$ is the label of the node $v$. In the main paper, we define multi-label node and edge homophily by,

$$h^{\text{edge, ml}} = \frac{1}{|\mathbb{A}|} \sum_{(u,v) \in \mathbb{A}} \frac{|\mathbb{L}_u \cap \mathbb{L}_v|}{|\mathbb{L}_u \cup \mathbb{L}_v|}, \ h^{\text{node, ml}} = \frac{1}{|\mathbb{V}|} \sum_{v \in \mathbb{V}} \left( \frac{1}{|\mathcal{N}(v)|} \sum_{u \in \mathcal{N}(v)} \frac{|\mathbb{L}_u \cap \mathbb{L}_v|}{|\mathbb{L}_u \cup \mathbb{L}_v|} \right). \quad (45)$$

If we compute $r = \frac{|\mathbb{L}_u \cap \mathbb{L}_v|}{|\mathbb{L}_u \cup \mathbb{L}_v|}$ for single-label multi-class graphs, $r = \frac{1}{1} = 1$ for nodes of same classes, and $r = \frac{0}{2} = 0$ for nodes of different classes. That makes $h^{\text{edge, ml}} = h^{\text{edge}}$ and $h^{\text{node, ml}} = h^{\text{node}}$ for single-label graphs.

## A.8 PERFORMANCE OF DIFFERENT GNN LAYERS

In Table 6, we demonstrate the performance of S2N models using additional GNN layers: Graph Isomorphism Networks (GIN) (Xu et al., 2019) and Graph Attention Networks V2 (GATv2) (Brody et al., 2022).

GIN and GATv2 (S2N+0 and S2N+A) outperform GLASS on PPI-BPbut perform worse than GLASS on EM-User. We confirm that S2N outperforms classic data structures: separated and connected forms. For GATv2, we cannot experiment with the connected form on EM-User due to the requirements of large GPU memory. Nonetheless, all S2N models with GIN and GATv2 outperform SubGNN on all datasets.

Compared to GCNII, which showed the best performance in our paper, GIN and GATv2 generally perform worse. This implies that architectures designed for node or link-level tasks are sub-optimal for subgraph-level tasks. We suggest further studies on model architectures for learning subgraph representations.

## A.9 ABLATION STUDY OF HYPERPARAMETERS

We conduct ablation studies on the readout method (Equation 8) (sum, mean, max, and degree-dependent) and the number of layers in $\text{GNN}_{\text{S2N}}$ (0, 1, 2, 4). We report the performance of S2N+0 and S2N+A with GCNII by the readout method in Table 7 and the number of layers in Table 8.

---

[4]https://pytorch.org/docs/1.9.0/generated/torch.cuda.max_memory_allocated.html

Table 6: Mean performance in micro F1-score over 10 runs. We mark with daggers the reprinted results from Alsentzer et al. (2020) (†) and Wang & Zhang (2022) (‡).

| Model | Data Structure | PPI-BP | EM-User |
|---|---|---|---|
| Sub2Vec Best[†] | | $30.9_{\pm 2.3}$ | $85.9_{\pm 1.4}$ |
| SubGNN[†] | | $59.9_{\pm 2.4}$ | $81.4_{\pm 4.6}$ |
| GLASS[‡] | | $61.9_{\pm 0.7}$ | $88.8_{\pm 0.6}$ |
| GCNII | Separated | $61.3_{\pm 1.2}$ | $84.7_{\pm 4.1}$ |
| GCNII | Connected | $63.5_{\pm 2.0}$ | $85.5_{\pm 4.8}$ |
| GCNII | S2N+0 | $63.5_{\pm 2.4}$ | $86.5_{\pm 3.2}$ |
| GCNII | S2N+A | $63.7_{\pm 2.3}$ | $89.0_{\pm 1.6}$ |
| GIN | Separated | $60.6_{\pm 2.1}$ | $82.2_{\pm 6.6}$ |
| GIN | Connected | $61.0_{\pm 3.3}$ | $83.7_{\pm 4.8}$ |
| GIN | S2N+0 | $63.3_{\pm 1.6}$ | $84.9_{\pm 5.3}$ |
| GIN | S2N+A | $62.2_{\pm 1.9}$ | $83.1_{\pm 1.6}$ |
| GATv2 | Separated | $61.4_{\pm 2.6}$ | $84.7_{\pm 4.9}$ |
| GATv2 | Connected | $61.0_{\pm 1.5}$ | OOM |
| GATv2 | S2N+0 | $62.8_{\pm 1.7}$ | $84.9_{\pm 2.4}$ |
| GATv2 | S2N+A | $62.6_{\pm 1.4}$ | $86.7_{\pm 3.2}$ |

Table 7: Mean performance in micro F1-score over 10 runs using GCNII models with different readout methods.

| Data Structure | Readout | PPI-BP | HPO-Neuro | HPO-Metab | EM-User |
|---|---|---|---|---|---|
| S2N+0 | Sum | $\mathbf{63.5}_{\pm \mathbf{2.4}}$ | $\mathbf{66.4}_{\pm \mathbf{1.1}}$ | $61.6_{\pm 1.7}$ | $\mathbf{86.5}_{\pm \mathbf{3.2}}$ |
| | Mean | $59.9_{\pm 2.0}$ | $63.9_{\pm 0.5}$ | $\mathbf{62.0}_{\pm \mathbf{1.0}}$ | $85.3_{\pm 3.9}$ |
| | Max | $48.6_{\pm 2.8}$ | $54.1_{\pm 1.0}$ | $51.8_{\pm 2.2}$ | $72.2_{\pm 7.1}$ |
| | Degree | $60.3_{\pm 2.2}$ | $65.6_{\pm 1.0}$ | $58.8_{\pm 2.3}$ | $83.1_{\pm 2.9}$ |
| S2N+A | Sum | $\mathbf{63.7}_{\pm \mathbf{2.3}}$ | $\mathbf{68.4}_{\pm \mathbf{1.0}}$ | $\mathbf{63.2}_{\pm \mathbf{2.7}}$ | $88.8_{\pm 2.1}$ |
| | Mean | $59.6_{\pm 1.5}$ | $66.6_{\pm 1.0}$ | $60.8_{\pm 1.1}$ | $88.0_{\pm 3.2}$ |
| | Max | $58.5_{\pm 1.7}$ | $59.6_{\pm 2.0}$ | $59.4_{\pm 2.7}$ | $81.2_{\pm 3.1}$ |
| | Degree | $63.1_{\pm 2.2}$ | $68.4_{\pm 0.9}$ | $61.0_{\pm 2.0}$ | $\mathbf{89.0}_{\pm \mathbf{1.6}}$ |

Generally, the sum-readout performs best, and the max-readout performs the worst, as illustrated in Table 7. The performance of mean-readout and degree-dependent readout varies by dataset. In S2N+A, degree-dependent readout performs similarly to sum-readout and slightly outperforms on EM-User.

In Table 8, we find that using message-passing (i.e., the number of layers $> 0$) always increases the performance on all datasets. That is, modeling the S2N graph structures helps to learn the representation of subgraphs. The performance improvement by GNN$_{\text{S2N}}$ in S2N+0 is higher than in S2N+A, which leverages internal structures. The performance decreases when we use a deeper GNN$_{\text{S2N}}$ than the optimum; that is, an over-smoothing effect exists in GNN$_{\text{S2N}}$ (Li et al., 2018).

### A.10 PERFORMANCE AND EFFICIENCY OF COARSENED S2N IN A DATA-SCARCE SETTING

For experiments in a data-scare setting, we narrow the search space of hyperparameters. Specifically, we fix to use batch normalization but not skip-connections. We use a coarsening ratio that creates virtual subgraphs smaller than the average size: $[0.2, 0.3, 0.4, 0.5, 0.6]$ for PPI-BP and $[0.8, 0.9]$ for EM-User. We follow the same tuning procedures in Appendix A.5 for the remaining details.

As stated in §6.4, we summarize performance and efficiency on EM-User in Figure 6. Overall, results on EM-User do not show a notable difference from trends in data-scarce experiments on PPI-BP at §6.4. The difference is that while S2N+0 underperforms the baselines, S2N+A outperforms the rest by a large margin in all experiments. This suggests that the internal structures in subgraphs are essential to EM-User. The observation that the performance of connected and separated forms is similar also supports this. Nevertheless, the results of S2N+A confirm that message-passing between subgraphs improves representation quality.

Table 8: Mean performance in micro F1-score over 10 runs using GCNII models with different numbers of layers of $\text{GNN}_{\text{S2N}}$.

| Data Structure | # layers | PPI-BP | HPO-Neuro | HPO-Metab | EM-User |
|---|---|---|---|---|---|
| S2N+0 | 0 | $57.7_{\pm 1.6}$ | $65.2_{\pm 1.6}$ | $55.5_{\pm 1.9}$ | $77.6_{\pm 9.4}$ |
| | 1 | $61.1_{\pm 2.4}$ | $\mathbf{66.4_{\pm 1.1}}$ | $\mathbf{61.6_{\pm 1.7}}$ | $79.2_{\pm 9.2}$ |
| | 2 | $\mathbf{63.5_{\pm 2.4}}$ | $65.6_{\pm 1.4}$ | $59.4_{\pm 1.0}$ | $\mathbf{86.5_{\pm 3.2}}$ |
| | 4 | $62.8_{\pm 2.0}$ | $65.8_{\pm 0.8}$ | $61.1_{\pm 1.7}$ | $79.2_{\pm 7.9}$ |
| S2N+A | 0 | $59.7_{\pm 2.2}$ | $68.2_{\pm 0.8}$ | $61.8_{\pm 1.7}$ | $87.1_{\pm 3.5}$ |
| | 1 | $\mathbf{63.7_{\pm 2.3}}$ | $\mathbf{68.4_{\pm 1.0}}$ | $61.9_{\pm 2.0}$ | $\mathbf{89.0_{\pm 1.6}}$ |
| | 2 | $61.8_{\pm 1.4}$ | $68.0_{\pm 0.8}$ | $\mathbf{63.2_{\pm 2.7}}$ | $86.3_{\pm 4.9}$ |
| | 4 | $61.6_{\pm 1.7}$ | $67.7_{\pm 0.8}$ | $62.0_{\pm 1.6}$ | $86.3_{\pm 5.2}$ |

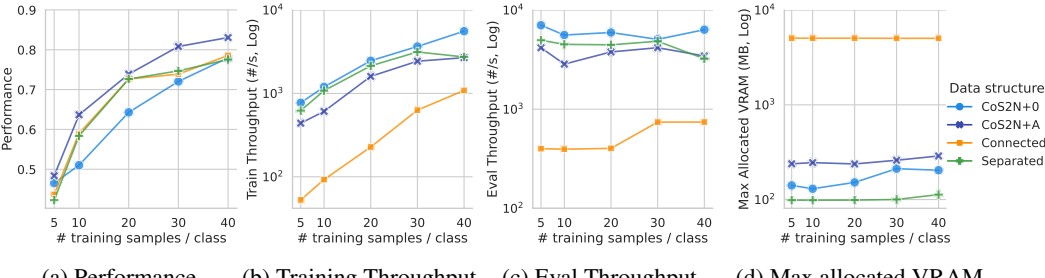

(a) Performance    (b) Training Throughput    (c) Eval Throughput    (d) Max allocated VRAM

Figure 6: Performance and efficiency on EM-User of CoS2N, connected, and separated forms by the number of training samples in a data-scarce setting.

We report the performance on PPI-BP and EM-User with respect to the coarsening ratio in Figure 7. Although there are differences depending on the number of training samples, we can conclude that finding the optimal coarsening ratio for each dataset can increase the performance. In addition, when the training samples are too small, the difference between S2N+0 and S2N+A by the coarsening ratio change is not noticeable.

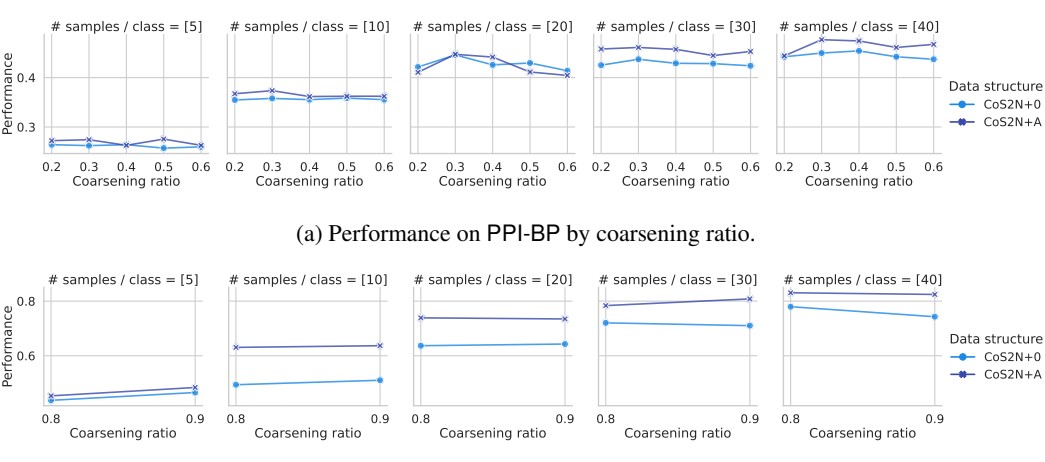

(a) Performance on PPI-BP by coarsening ratio.

(b) Performance on EM-User by coarsening ratio.

Figure 7: Performance of CoS2N on PPI-BP and EM-User by coarsening ratio.

