# OpenReview forum: "Subgraph-To-Node Translation for Efficient Representation Learning of Subgraphs"
_ICLR.cc/2024/Conference — Submitted to ICLR 2024_

### Official Review · Reviewer_1NPi · 2023-11-01

**Soundness:** 2 fair
**Presentation:** 2 fair
**Contribution:** 2 fair
**Rating:** 5
**Confidence:** 3

**Summary:**

This paper looks at the problem of supervised subgraph classification. To handle the scalability issues with the existing models, the authors propose the Subgraph-To-Node (S2N) translation, an efficient data structuring mechanism for manipulating subgraphs prior to model design. They also explore graph coarsening techniques in this context in a data-scarce setting. The authors prove that the S2N node representations approximate the subgraph representations of the original global graph. Their experiments are designed to show that S2N substantially reduces memory and time costs with little degradation in performance.

**Strengths:**

The exposition is quite clear. The proposed solutions are quite straighforward and easy to follow. The experimental protocol is more or less quite detailed and sufficiently tests the proposed models. The idea of using graph coarsening for subgraph classification is novel (to the best of my knowledge) and deserves further study.

**Weaknesses:**

1. There are not many real-world datasets for the supervised subgraph classification. The authors should definitely consider synthetic datasets, e.g. those considered by Alsenter et. al. (2020). It is not clear why the authors have not considered such synthetic datasets.

2. The authors do not clarify if the considered datasets are adequately large so that the GPU speed/memory really forms a bottleneck for learning.

3. I am not sure about handpicking the Configuration Model (CM) as a justification for low computational complexity of S2N. Why has this model been picked: One could definitely study some other random graph models and ask the same questions?

4. When you coarsen a graph, how is the structure of the original subgraphs preserved? It is not clear how to use the coarsened graph to say something about the subgraphs in the original graph.

Overall, the paper is not substantial enough for significant research impact.

**Questions:**

Please comment on the enumerated points in Weaknesses above.

---

> ### Author Response · Authors · 2023-11-17
> **Response to `1NPi` (1)**
>
> Thank you for your review and constructive feedback. We provide answers to your questions below.
>
> ## More Experiments on Synthetic Datasets
> > There are not many real-world datasets for the supervised subgraph classification. The authors should definitely consider synthetic datasets, e.g. those considered by Alsenter et. al. (2020). It is not clear why the authors have not considered such synthetic datasets.
>
> Thank you for your suggestion. We agree with your comment about synthetic datasets and have conducted experiments on them. The table below summarizes the performance of GCNII+S2N on synthetic datasets.
>
> |     | Density    | Cut-Ratio  | Coreness   | Component   |
> |----------------------|------------|------------|------------|-------------|
> | GLASS        | 93.0 ± 0.9 | 93.5 ± 0.6 | 84.0 ± 0.9 | 100.0 ± 0.0 |
> | SubGNN               | 91.9 ± 1.6 | 62.9 ± 3.9 | 65.9 ± 9.2 | 95.8 ± 9.8  |
> | Sub2Vec              | 45.9 ± 1.2 | 35.4 ± 1.4 | 36.0 ± 1.9 | 65.7 ± 1.7  |
> | GCNII / S2N+0        | 67.2 ± 2.4 | 56.0 ± 0.0 | 57.0 ± 4.9 | 100.0 ± 0.0 |
> | GCNII / S2N+A        | 93.2 ± 2.6 | 56.0 ± 0.0 | 85.7 ± 5.8 | 100.0 ± 0.0 |
> | GCNII / S2N+0 + RWPE | 74.8 ± 3.6 | 85.2 ± 5.1 | 56.1 ± 3.0 | 100.0 ± 0.0 |
> | GCNII / S2N+0 + RWPE | 93.6 ± 2.0 | 89.2 ± 2.6 | 77.4 ± 9.1 | 100.0 ± 0.0 |
>
> S2N+A outperforms the state-of-the-art (GLASS) on Density, Coreness, and Component datasets. S2N+0 shows the same performance as GLASS only in Component. For each dataset, the attributes that affect the subgraph properties (i.e., labels of synthetic datasets) are known (as illustrated in the table below) [1]. Because S2N compresses the global graph structure, it is challenging to learn Cut-Ratio, which requires exact information about the border structure (i.e., global structure). Learning Density and Coreness of subgraphs require their internal structures. Therefore, S2N+0, which does not maintain internal structure, relatively underperforms baselines.
>
> | Density | Cut-Ratio  | Coreness  | Component     |
> |--------------|---------|--------------------|-----------------|
> | Internal structure | Border structure | Internal structure, border structure and position | Internal and external position |
>
> To address this issue, we can add structural encoding to the input features, particularly Random Walk Positional Encoding (RWPE) [2]. The efficiency of S2N is maintained since the RWPE is computed once before training and only requires the memory complexity of $O(N)$. RWPE allows S2N to significantly improve the performance of Density and CutRatio, but not of Coreness. We interpret that RWPE for subgraphs can encode internal and border structures well but cannot encode border positions. We leave the development of structural encoding for S2N as future work.
>
> Above all, tasks that depend only on specific attributes barely exist in the real world. We can find evidence from the high performance of S2N on real-world datasets. Our proposed methods provide GNN practitioners with an efficient solution for real-world subgraph-level applications. It is analogous to lossy compression of images (e.g., JPEG) to serve them efficiency-sensitive applications.
>
> We will add all results and discussion to the revised paper.
>
> [1] Alsentzer, Emily, et al. "Subgraph neural networks." Advances in Neural Information Processing Systems 33 (2020): 8017-8029.
>
> [2] Dwivedi, Vijay Prakash, et al. "Graph Neural Networks with Learnable Structural and Positional Representations." International Conference on Learning Representations. 2022.
>
> ## Clarification on Computational & Memory Bottleneck
> > The authors do not clarify if the considered datasets are adequately large so that the GPU speed/memory really forms a bottleneck for learning.
>
> For the datasets we used, the number of nodes is 17K, 15K, 57K, and the number of edges is 317K, 3.2M, 4.6M (Table 4 in the Appendix). In the table below, we introduce some datasets referred to as large-scale in our GNN research community. The number of edges compared to these datasets is at a similar level; thus, similar scalability is required to model the datasets in this paper using GNNs. Particularly, when using attention models (e.g., GATv2), we cannot load the largest dataset using baselines into a single GPU of 11G VRAM.
>
> | Name          | # Nodes | # Edges | Source |
> |---------------|---------|---------|--------|
> | Penn94        | 42K     | 1.4M    | [1]    |
> | arXiv-year    | 169K    | 1.2M    | [1]    |
> | twitch-gamers | 168K    | 6.8M    | [1]    |
> | ogbl-ddi      | 4K      | 1.3M    | [2]    |
> | ogbl-biokg    | 93K     | 5.4M    | [2]    |
> | ogbn-arxiv    | 169K    | 1.2M    | [2]    |
>
> [1] Lim, Derek, et al. "Large scale learning on non-homophilous graphs: New benchmarks and strong simple methods." Advances in Neural Information Processing Systems 34 (2021): 20887-20902.
>
> [2] Hu, Weihua, et al. "Open graph benchmark: Datasets for machine learning on graphs." Advances in neural information processing systems 33 (2020): 22118-22133.

---

> ### Author Response · Authors · 2023-11-17
> **Response to `1NPi` (2)**
>
> ## Justification for the Choice of the Random Graph Model
> > I am not sure about handpicking the Configuration Model (CM) as a justification for low computational complexity of S2N. Why has this model been picked: One could definitely study some other random graph models and ask the same questions?
>
> The complexity of S2N strongly depends on the distribution of translated edge weights. Thus, we need a random graph model that can analytically calculate the distribution of edge weights (i.e., the number of shared edges in two subgraphs). When using the configuration model (CM), the distribution of S2N's edge weights can be derived from the degree distribution of the global graph. This is possible because CM calculates the probability of edge existence through the degrees of a pair of nodes. Note that CM is frequently used in analytically calculating numerous network measures [1].
>
> We also emphasize that CM only requires a degree sequence or a distribution. That means CM can also generate graphs generated by other random graph models. For example, when the degree distribution is Poisson distribution, CM generates graphs close to the Erdős–Rényi model. CM can also generate degree distributions with other distributions, for example, power-law distributions. See [2] for more details.
>
> We will add this justification to the revised paper.
>
> [1] Barabási, Albert-László. "Network science." Philosophical Transactions of the Royal Society A: Mathematical, Physical and Engineering Sciences 371.1987 (2013): 20120375.
>
> [2] Newman, Mark. Networks. Oxford university press, 2018.
>
> ## Answers to Questions
> > When you coarsen a graph, how is the structure of the original subgraphs preserved? It is not clear how to use the coarsened graph to say something about the subgraphs in the original graph.
>
> It seems that graph coarsening in CoS2N has not been written clearly and in detail. The details of this are as follows, and we will include them in the revised paper.
>
> 1. Apply the graph coarsening method to the global graph. The output is a partition of nodes in the global graph. In other words, graph coarsening assigns one super-node to each node in the global graph.
> 2. Construct induced subgraphs of the global graph per a set of nodes in the same super-node. We call them virtual subgraphs.
> 3. Merge a set of virtual subgraphs and an existing set of real subgraphs. Until this step, the structures of the original subgraph are preserved as is.
> 4. Perform S2N translation of this union set of subgraphs.

---

> > ### Comment · Reviewer_1NPi · 2023-11-19
> >
> > Thank you for your answers!

---

> > > ### Author Response · Authors · 2023-11-20
> > > **Thank you for your reply (1NPi)**
> > >
> > > Thank you for your acknowledgment of our author response.
> > >
> > > We have uploaded [the revised paper](https://openreview.net/pdf?id=BeuTCoe3bf). The changed or added part is highlighted with the red text. We believe that your comments are helpful and the revision addressed your concerns and questions. If the reviewer agrees, we would appreciate it if you consider raising the score. If you still have concerns or questions, we would love to discuss them.

---

> > > ### Author Response · Authors · 2023-11-22
> > > **Dear Reviewer 1NPi**
> > >
> > > Dear Reviewer 1NPi.
> > >
> > > We kindly remind you that the end of the discussion stage is now approaching. We uploaded [the revised paper](https://openreview.net/pdf?id=BeuTCoe3bf) addressing your concerns. If you have any additional questions after reading our revised paper, we are happy to discuss them.
> > >
> > > In summary, our revision includes:
> > > 1. We added and discussed results on four synthetic datasets for the supervised subgraph classification (Table 2 and Section 6.2).
> > > 2. We clarified that the used datasets are adequately large and that scalability is needed for real-world subgraph-level tasks (Appendix A.4).
> > > 3. We justified using the Configuration Model (CM) to analyze the complexity of S2N (Section 4.1 and Appendix A.2).
> > > 4. We added more details on how the structure of the original subgraphs is preserved in graph coarsening (Section 3.4).
> > >
> > > Thank you again for your time and effort in reviewing our paper. We are looking forward to your response.

---

### Official Review · Reviewer_tdbM · 2023-11-01

**Soundness:** 3 good
**Presentation:** 3 good
**Contribution:** 2 fair
**Rating:** 5
**Confidence:** 3

**Summary:**

This paper proposes Subgraph-to-node (S2N), an efficient data structure for subgraph-level prediction. The nodes of the new structure correspond to the subgraph and the edges are the relations among the subgraphs. The results shows both high efficiency and performance.

**Strengths:**

1. The proposed method shows high performance compared to the existing data structures and other baselines even with a simple and straightforward method.
2. The clear figures help in understanding and the presentation of the proposed work.

**Weaknesses:**

1. The graph coarsening process lacks novelty. It is quite straightforward and well-known to treat the subgraphs into a single node and link the nodes that share the nodes in the original graphs.
2. Can this approach distinguish the two subgraphs that share the same number of nodes, i.e., is the proposed structure reconstructable? For instance, what if the red subgraph is connected to the right side of the blue subgraph in Figure 1? It may generate the same subgraphs and same number of shared nodes.
3. Lack of details for the selection step of the subgraphs to be mapped into new nodes. How do you select the subgraphs and how do you prove that the selected subgraph is the optimal choice?
4. Lack of backbone architectures, which are limited to GCN-based. What about other GNN backbone architectures such as GIN? Is the proposed method restricted only to GCN as proved in Section 4.2?

**Questions:**

1. What is the difference between the existing works and the proposed works on super-nodes? I cannot clearly understand what the node boundaries in super-nodes are unknown is in Section 2.

---

> ### Author Response · Authors · 2023-11-17
> **Response to `tdbM`**
>
> Thank you for your review and constructive feedback. We address your comment on the lack of backbone architectures in the [general response](https://openreview.net/forum?id=BeuTCoe3bf&noteId=6ld2E20A6l). Here, we provide answers to your individual questions below.
>
> ## ​​Lack of Novelty
> > The graph coarsening process lacks novelty. It is quite straightforward and well-known to treat the subgraphs into a single node and link the nodes that share the nodes in the original graphs.
>
> We argue that the value of simplicity and the improved performance and efficiency over existing methods reinforce the genuine novelty of our contributions. We would like to offer the following points to counter the critique.
> - **Value of Simplicity**: The simplicity of our proposed approach is one of its key strengths and contributions. By distilling the problem to its core and focusing on a simple yet effective idea, we can provide an efficient and effective solution for subgraph-level tasks.
> - **Novel Perspective and Fundamental Question**: We emphasize that it is not well-known to treat the subgraphs into a single node and link the nodes that share the nodes for subgraph representation learning. How to design an efficient data structure for subgraphs is a novel perspective that not only complements but advances the field's understanding and approaches to the problem.
> - **Outperforming State-of-the-Art**: We note that our proposed S2N translation is on par with or better than the state-of-the-art. Our approach is fast and lightweight while maintaining performance levels, and should not be dismissed as straightforward.
>
> Last but not least, we would like to quote the other reviewer `1NPi`'s evaluation of the novelty: *"The idea of using graph coarsening for subgraph classification is novel (to the best of my knowledge) and deserves further study."*
>
> We think our strengths have not clearly revealed in the introduction, and based on the above bullets, we will revise them so that they are clearly visible. We would appreciate you reconsidering the novelty of our method.
>
> ## Answers to Questions
> > Can this approach distinguish the two subgraphs that share the same number of nodes, i.e., is the proposed structure reconstructable? For instance, what if the red subgraph is connected to the right side of the blue subgraph in Figure 1? It may generate the same subgraphs and same number of shared nodes.
>
> Generally, S2N can distinguish between two subgraphs with the same number of nodes. This is because S2N's node will have different neighborhoods in the S2N graph depending on neighbor subgraphs in the global graph. The case where S2N cannot distinguish between two subgraphs of the same size is when (1) the global structure around subgraphs is completely identical and (2) features of subgraphs are identical. However, it is difficult to exist due to real-world graphs' rich features and complex dynamics.
>
> > Lack of details for the selection step of the subgraphs to be mapped into new nodes. How do you select the subgraphs and how do you prove that the selected subgraph is the optimal choice?
>
> In this study, all given subgraphs are mapped to new nodes. In detail, all training subgraphs are translated into nodes in the training stage, and all training and evaluation subgraphs are translated into nodes in the evaluation stage (See Section 5). How to select subgraphs that make up the optimal S2N graph is an interesting research topic but is not the scope of this paper. We leave this as a future study and will mention it in the revised paper.
>
> > What is the difference between the existing works and the proposed works on super-nodes? I cannot clearly understand what the node boundaries in super-nodes are unknown is in Section 2.
>
> A node boundary is the boundary between a subgraph (or super-nodes) and the global graph. When we say node boundary is unknown, the super-node is not given to existing graph coarsening methods. These methods aim to find a set of super-nodes that partitions a given graph according to specified constraints. Unlike these, we view given subgraphs as super-nodes and explore whether performance and efficiency differ in subgraph-level downstream tasks when the super-nodes are mapped to nodes. That is, the purpose and primary conditions of the research are different. We will revise this part clearly by choosing straightforward terminologies.

---

> > ### Comment · Reviewer_tdbM · 2023-11-20
> >
> > Thank you for the kind response and I slightly revised the score. However, I still have some remaining concerns.
> >
> > I first don’t agree that it is not well-known to treat the subgraphs into a single node and link the nodes that share the nodes. For instance, a popular graph generative model, JT-VAE, proposes to decompose the graph into junction tree, where a single node corresponds to the motif and edges link the nodes that share the nodes. It might not be well-known for subgraph representation learning, the concept of mapping subgraphs to a single node and linking the nodes is quite well-known for other graph neural network area such as graph generative models. However, I now understand that the idea of using graph coarsening for subgraph representation learning could be novel as the reviewer 1NPi mentioned.
> >
> > In addition, the empirical results using GIN and GAT given by the authors lack the details for the results with conventional data structure: separated and connected. Don’t we have to compare the GIN/GAT + S2N results with GIN/GAT + separated / connected, not GLASS and SubGNN to observe the enhancement of S2N data structure?
> >
> > Thank you for the detailed response again.
> >
> > [1] Jin, W., Barzilay, R., & Jaakkola, T. (2018, July). Junction tree variational autoencoder for molecular graph generation. In International conference on machine learning (pp. 2323-2332). PMLR.

---

> > > ### Author Response · Authors · 2023-11-20
> > > **Thank you for your reply (tdbM)**
> > >
> > > Thank you for your reply. We are happy to engage in further discussion.
> > >
> > > > Thank you for the kind response and I slightly revised the score.
> > >
> > > Thank you for revising your initial decision. However, my observation shows that the rating has not been changed yet. I would appreciate it if you could double-check to ensure nothing has been missed.
> > >
> > > > I first don’t agree that it is not well-known to treat the subgraphs into a single node and link the nodes that share the nodes. For instance, a popular graph generative model, JT-VAE, proposes to decompose the graph into junction tree, where a single node corresponds to the motif and edges link the nodes that share the nodes. It might not be well-known for subgraph representation learning, the concept of mapping subgraphs to a single node and linking the nodes is quite well-known for other graph neural network area such as graph generative models. However, I now understand that the idea of using graph coarsening for subgraph representation learning could be novel as the reviewer 1NPi mentioned.
> > >
> > > First, thank you for understanding the novelty of our research from the perspective of graph coarsening for subgraph representation learning.
> > >
> > > We agree with the reviewer's comment regarding other existing tasks that map subgraphs to nodes (such as graph generative models). However, as the reviewer pointed out, this approach has not been used in the subgraph-level task, and rather, research for this task has been conducted by designing complex models. We emphasize that our primary contribution is not developing a technical and specialized method but exploring the fundamental question of subgraph representation learning and proposing a novel perspective.
> > >
> > > Nevertheless, we will survey existing studies of mapping subgraphs to nodes for various downstream tasks like JT-VAE. In the next revision, we will include this line of work and compare it with our proposed S2N.
> > >
> > > > In addition, the empirical results using GIN and GAT given by the authors lack the details for the results with conventional data structure: separated and connected. Don’t we have to compare the GIN/GAT + S2N results with GIN/GAT + separated / connected, not GLASS and SubGNN to observe the enhancement of the S2N data structure?
> > >
> > > This is a valid critique and thank you for pointing it out. We are now conducting experiments using GIN/GAT plus separated/connected forms. We will upload the results as they come out. Please understand that getting full results before the discussion period might not be possible since the connected form requires more computational resources.

---

> > > > ### Comment · Reviewer_tdbM · 2023-11-20
> > > >
> > > > Sorry for the confusion. I revised my score just right before.

---

> > > > > ### Author Response · Authors · 2023-11-23
> > > > >
> > > > > Dear Reviewer tdbM
> > > > >
> > > > > We uploaded [the revised paper](https://openreview.net/pdf?id=BeuTCoe3bf) that addresses remaining issues:
> > > > > 1. In the related work section (Section 2), we added additional papers about mapping subgraphs to nodes to represent meaningful clusters. In particular, we compared our work with JT-VAE in Appendix A.1.3.
> > > > > 2. We updated results using GIN and GAT with separated and connected forms on PPI-BP and EM-User datasets (Table 6 and Appendix A.8). For GIN and GAT, We confirm that S2N outperforms classic data structures: separated and connected forms.
> > > > >
> > > > > Again, thank you for the detailed feedback, and we believe that your comments help strengthen our paper. If there is an issue that still needs to be addressed, we would be happy to discuss it. If reviewers think the current paper addresses all concerns, we would appreciate it if you consider raising the score.

---

### Official Review · Reviewer_AUEj · 2023-11-05

**Soundness:** 3 good
**Presentation:** 3 good
**Contribution:** 3 good
**Rating:** 6
**Confidence:** 4

**Summary:**

This paper proposes S2N and CoS2N, two new methods for learning the representation of subgraphs where the subgraphs are given as input to the model as well as the original whole graph. The proposed methods are simple and effective, as evidenced by superior results on four real-world datasets.

**Strengths:**

1. Both theoretical analysis and experimental support are provided to show the advantages of the proposed methods.
2. The model design is quite simple yet the results are impressive, both in terms of effectiveness and efficiency.

**Weaknesses:**

1. There lack of ablation study of certain hyperaprameters and design choices. For example, the authors "use two well-known GNNs" but it is unclear why alternative choices are not discussed or used. Given the abundance of GNN models nowadays and the fact that GCN and GCN2 are relatively earlier (before 2021), it is unclear if the adoption of more recent GNN models could yield better results. The authors mention one baseline, SubGNN, uses pre-trained embeddings by GIN, yet there is no explanation of why GIN can or cannot be used for the proposed methods.
What is more important, the number of layers is tuned between 1 and 2 layers (Section A.3), and it is unclear how much performance fluctuates with even more or less (0 layers, i.e. no message passing) layers. Similarly, it is unclear if alternative readout methods and graph coarsening methods are experimented with. Adding such additional experiments certainly require more work and resource, but would further help improve the soundness of the paper.
2. I suggest the authors provide more descriptions of existing methods, esp. SubGNN and GLASS. For example, if and what GNN models are used. There is some detail in Appendix A, but an additional section that focuses on the architectural comparison of all the methods would further enhance the clarity of the paper.
3. Writing issues, e.g. lack of citation of GIN.

**Questions:**

1. How is the proposed CoS2N related to DiffPool "Ying, Zhitao, et al. "Hierarchical graph representation learning with differentiable pooling." Advances in neural information processing systems 31 (2018)."? At a high level, both of them perform pooling and allow further message passing between the pooled clusters/subgraphs. DiffPool adopts a learnable/differentiable way to pool nodes, whereas the proposed method adopts Variation Edges for coarsening. Of course, the task is different, yet I would like to hear from the authors more about the model-architecture-level comparison. This would help readers better see the novelty of the proposed methods with respect to related work designed for different asks.

---

> ### Author Response · Authors · 2023-11-17
> **Response to `AUEj` (1)**
>
> Thank you for your review and constructive feedback. We addressed questions about the lack of GNN models other than GCN and GCNII in the [general response](https://openreview.net/forum?id=BeuTCoe3bf&noteId=6ld2E20A6l). We provide answers to your individual questions below.
>
> ## Ablation Studies and Hyperparameter Analysis
> > What is more important, the number of layers is tuned between 1 and 2 layers (Section A.3), and it is unclear how much performance fluctuates with even more or less (0 layers, i.e. no message passing) layers. Similarly, it is unclear if alternative readout methods and graph coarsening methods are experimented with. Adding such additional experiments certainly require more work and resource, but would further help improve the soundness of the paper.
>
> Thank you for your suggestion. Now, we are planning to conduct ablation studies on important hyperparameters, including the number of layers, and readouts. Due to the computational resource constraint, we have yet to finish the experiments, but we will include the results as soon as they come out.
>
> ## Detailed Descriptions of Architectural Differences
> > I suggest the authors provide more descriptions of existing methods, esp. SubGNN and GLASS. For example, if and what GNN models are used. There is some detail in Appendix A, but an additional section that focuses on the architectural comparison of all the methods would further enhance the clarity of the paper.
>
> We agree with the reviewer's comment that a detailed architecture description will help with clarity. We describe the architectural differences between SubGNN, GLASS, and S2N below. This will be included in the Appendix.
>
> SubGNN, GLASS, and S2N improve different parts of the machine learning pipeline to solve subgraph-level tasks. SubGNN designs a whole model, GLASS augments input data through a labeling trick, and S2N uses a new data structure.
>
> SubGNN performs message-passing between subgraphs (or patches). Through this, the properties of internal and border structures for three channels (position, neighborhood, and structure) are learned independently. To learn a total of 6 (2 x 3) properties, SubGNN designs patch samplers, patch representation, and similarity (weights of messages) for each property in an ad hoc manner. To learn internal positions, for example, SubGNN patches nodes inside the subgraph uses its representation as a message, and uses distance-based similarity as weights. By the complex model design, SubGNN requires a lot of computational resources for data pre-processing, model training, and inference.
>
> GLASS uses plain GNNs but labels input nodes as to whether they belong to the subgraph (the label of one) or not (the label of zero). Separate node-level message-passing is performed for each label to distinguish the internal and border structures of the subgraph. GLASS's labeling trick is effective, but hard to handle multiple labels from multiple subgraphs in a batch. Although the authors of GLASS propose a max-zero-one trick to address this issue, small batches are still recommended. In addition, using a large global graph requires significant computational and memory resources.
>
> In comparison, our proposed S2N uses the new data structure that stores and processes subgraphs efficiently. By compressing the global graph, computational and memory resource requirements are reduced. There are no restrictions on batch learning so we can train S2N graphs in the full batch.

---

> ### Author Response · Authors · 2023-11-17
> **Response to `AUEj` (2)**
>
> ## Answers to Questions
> > How is the proposed CoS2N related to DiffPool "Ying, Zhitao, et al. "Hierarchical graph representation learning with differentiable pooling." Advances in neural information processing systems 31 (2018)."? At a high level, both of them perform pooling and allow further message passing between the pooled clusters/subgraphs. DiffPool adopts a learnable/differentiable way to pool nodes, whereas the proposed method adopts Variation Edges for coarsening. Of course, the task is different, yet I would like to hear from the authors more about the model-architecture-level comparison. This would help readers better see the novelty of the proposed methods with respect to related work designed for different asks.
>
> Thanks for the suggestion on the new perspective. We provide a model-level comparison between CoS2N and DiffPool below. This will be added to the Appendix.
>
> DiffPool learns the hierarchy of a graph to obtain graph-level representations. DiffPool softly assigns each node to a cluster during training by optimizing the downstream task loss. To stabilize the soft clustering assignment, the authors of DiffPool employ link prediction loss and entropy regularization loss. The problem is that the assignment matrix must be maintained in GPU memory, which requires quadratic memory complexity regarding the number of nodes. In other words, we cannot apply DiffPool to large graphs such as global graphs in our use cases.
>
> This study aims to efficiently perform subgraph representation learning by compressing data to be loaded into GPU memory. Memory-intensive graph coarsening, such as DiffPool's soft clustering assignment, should not be used to keep CoS2N efficient. Instead, we can secure the efficiency of CoS2N by performing graph coarsening before training the model, relying only on the structure of the global graph.

---

> ### Author Response · Authors · 2023-11-22
> **Dear Reviewer AUEj**
>
> Dear Reviewer AUEj.
>
> We kindly remind you that there is one day left until the discussion period ends. We uploaded [the revised paper](https://openreview.net/pdf?id=BeuTCoe3bf) addressing your concerns. If you have any additional questions after reading our revised paper, we are happy to discuss them.
>
> In summary, our revision includes:
> 1. We added experiments using GATv2 (a more recent GNN) and GIN (Table 6 and Appendix A.8).
> 2. We cited the GIN paper.
> 3. We added ablation studies regarding the number of layers (including no message-passing) and readout methods (Table 7, 8 and Appendix A.9).
> 4. We provided more descriptions of existing methods, SubGNN and GLASS (Appendix A.1.1).
> 5. We added the model-architecture-level comparison with DiffPool (Appendix A.1.2).
>
> Thank you again for your time and effort in reviewing our paper. We are looking forward to your response.

---

### Author Response · Authors · 2023-11-17
**General Response**

We appreciate the constructive feedback and efforts of reviewers. In this general response, we answer the issue raised by multiple reviewers: lack of backbone GNN architectures.

We first emphasize that the GNN layer is agnostic to S2N models. S2N translation does not affect model selection; thus, any GNN layers (including GIN [1]) can take S2N-translated graphs as inputs. In this paper, we want to demonstrate that even simple GNNs (GCN or GCNII) without complex operations can encode node representations by S2N. Note that GCN is the most widely used pioneer GNN model, and GCNII is its simple improvement.

Nevertheless, we agree with the reviewers that trying a diverse set of backbone models can strengthen our paper. So, we conduct additional experiments and analyses as follows and will include them in the next revision.

## Added Empirical Results Using GIN and GATv2

We demonstrate empirical results using GIN [1], pointed out by reviewers `tdbM` and `AUEj`. We also conduct experiments using GATv2 [2] to answer Reviewer `AUEj`’s question of whether adopting more recent GNN models could yield better results. Plus, our revised paper now cites GIN [1] (`AUEj`).

| Model  | Data Structure | PPI-BP     | HPO-Neuro  | HPO-Metab  | EM-User    |
|--------|----------------|------------|------------|------------|------------|
| SubGNN |                | 59.9 ± 2.4 | 63.2 ± 1.0 | 53.7 ± 2.3 | 81.4 ± 4.6 |
| GLASS  |                | 61.9 ± 0.7 | 68.5 ± 0.5 | 61.4 ± 0.5 | 88.8 ± 0.6 |
| GCNII  | S2N+0          | 63.5 ± 2.4 | 66.4 ± 1.1 | 61.6 ± 1.7 | 86.5 ± 3.2 |
| GCNII  | S2N+A          | 63.7 ± 2.3 | 68.4 ± 1.0 | 63.2 ± 2.7 | 89.0 ± 1.6 |
| GIN    | S2N+0          | 63.3 ± 1.6 | 67.1 ± 0.5 | 60.9 ± 1.4 | 84.9 ± 5.3 |
| GIN    | S2N+A          | 62.2 ± 1.9 | 67.1 ± 1.2 | 58.5 ± 3.0 | 83.1 ± 1.6 |
| GATv2  | S2N+0          | 62.8 ± 1.7 | 66.4 ± 0.7 | 61.1 ± 1.8 | 84.9 ± 2.4 |
| GATv2  | S2N+A          | 62.6 ± 1.4 | 64.5 ± 1.5 | 62.1 ± 1.6 | 86.7 ± 3.2 |

Both GIN and GATv2 (S2N+0 and S2N+A) outperform GLASS on PPI-BP. For HPO-Metab, GATv2 shows performance similar to GLASS, and its combination with S2N+A slightly outperforms GLASS. However, GIN and GATv2 perform worse than GLASS on HPO-Neuro and EM-User. Nonetheless, all S2N models with GIN and GATv2 outperform SubGNN on all datasets.

Compared to GCNII, which showed the best performance in our paper, GIN and GATv2 generally perform worse. This implies that architectures designed for node or link-level tasks are sub-optimal for subgraph-level tasks. We suggest further studies on model architectures for learning subgraph representations.

## Added Theoretical Results Using GIN

Reviewer `tdbM` questioned whether the theoretical results (Section 4.2) on the approximation error bound are generalizable to other GNNs. Note that Section 4.2 is not limited to GCN in its applicability. Intuitively, when sufficient subgraph samples are not available, message-passing in any GNNs fails in the global graph not covered by existing subgraphs. Analytically calculating this intuition into the error bound between S2N and the original graph strongly depends on the GNN design. Section 4.2 realizes this for GCN. We can obtain theoretical results similar to Section 4.2 for other GNNs, and each GNN layer requires separate hand-crafted analysis.

However, we might not get the approximation bound analytically depending on GNN architectures. For GIN as an example, the non-linearity in multi-layer perceptron (MLP) makes it hard to analytically compare the GIN outputs of S2N and the original graph. Instead, we introduce an approximation error bound on ‘GIN Sum-1-Layer’, a less powerful variant of GINs that replaces MLP with a single-layer perceptron. Note that GIN Sum-1-Layer is used to pre-train node embeddings in SubGNN [3]. See Proposition 4 and Corollary 1 in the Appendix of the revised paper.

## References

- [1] Xu, Keyulu, et al. "How Powerful are Graph Neural Networks?." International Conference on Learning Representations. 2018.
- [2] Brody, Shaked, Uri Alon, and Eran Yahav. "How attentive are graph attention networks?." ICLR (2021).
- [3] https://github.com/mims-harvard/SubGNN/blob/main/prepare_dataset/model.py#L21-L24

---

### Author Response · Authors · 2023-11-23
**Summary of Revision**

To all reviewers, thank you for all your efforts. We kindly remind you that only a few hours are left until the end of the discussion period. If there is an issue that still needs to be addressed, we would be happy to discuss it before the end of Nov 22nd (AOE).

We uploaded [the revised paper](https://openreview.net/pdf?id=BeuTCoe3bf) based on the reviewers' feedback. The revised parts are highlighted in red. We summarize the modified parts below:

- [`tdbM`] We revised the introduction section so that our novelty is clearly visible (Section 1)
- [`tdbM`] We revised the related work section to clarify the difference between existing works and methods (Section 2). Plus, we included the additional papers about mapping subgraphs to nodes for representing meaningful clusters in the related work section (Section 2). We added a detailed comparison with JT-VAE (Appendix A.1.3).
- [`1NPI`] We added more details on how the structure of the original subgraphs is preserved in graph coarsening (Section 3.4).
- [`1NPI`] We justified using the Configuration Model (CM) to analyze the complexity of S2N (Section 4.1 and Appendix A.2).
- [`1NPI`] We added and discussed results on four synthetic datasets for the supervised subgraph classification (Table 2 and Section 6.2).
- [`tdbM`] We added future work on selecting subgraphs that make up the optimal S2N graph (Section 7).
- [`AUEj` ] We provided more descriptions of existing methods, SubGNN, and GLASS (Appendix A.1.1). We added the model-architecture-level comparison with DiffPool (Appendix A.1.2)
- [`tdbM`] We added a theoretical analysis on an approximation error bound on a GIN variant (Appendix A.3).
- [`1NPI`] We clarified that the used datasets are adequately large and that scalability is needed for real-world subgraph-level tasks (Appendix A.4).
- [`AUEj` and `tdbM`] We added experiments using GATv2 (a more recent GNN) and GIN (Table 6 and Appendix A.8).
- [`AUEj`] We added ablation studies regarding the number of layers (including no message-passing) and readout methods (Table 7, 8, and Appendix A.9).

---

> ### Author Response · Authors · 2023-11-23
> **Dear All Reviewers and Area Chair**
>
> We would like to inform you that the discussion period will be closed soon. We have faithfully reflected on the reviewers' comments and questions in our latest revision. We would appreciate it if reviewers could respond to our responses and the revision.
>
> We sincerely thank you for your time and effort in reviewing our paper and for your constructive comments.

---

### Meta-Review · Area_Chair_m8kV · 2023-12-15

**Metareview:**

While the paper is commended for its clear writing and straightforward proposed methods, there are notable shortcomings that raise concerns. The experimental protocol is considered detailed, but the absence of exploration of synthetic datasets, particularly Alsenter et al. (2020), is a notable gap. The choice of the Configuration Model (CM) as a basis for asserting the low computational complexity of S2N is questionable and lacks justification. There is a missed opportunity to explore other random graph models and pose similar questions. Additionally, the paper lacks clarity on how the structure of the original subgraphs is preserved when coarsening a graph.

The paper is also critiqued for its lack of an ablation study on crucial hyperparameters and design choices. While the authors opt for two established GNNs, the absence of comparisons to other GNN models raises concerns about the relevance and potential performance improvements achievable with more recent architectures. The paper mentions the use of pre-trained embeddings by GIN in the SubGNN baseline but fails to provide a rationale for why GIN is selected or omitted for the proposed methods.

Furthermore, the tuning of the number of layers between 1 and 2 without exploring the impact of additional layers or considering scenarios with no message passing (0 layers) leaves uncertainties about the sensitivity of the proposed approach to layer variations. The lack of experimentation with alternative readout methods and graph coarsening techniques is noted as a limitation.

**Justification For Why Not Higher Score:**

Lack of experiments such as comparison to more recent GNN architectures.

**Justification For Why Not Lower Score:**

-

---

### Decision · Program_Chairs · 2024-01-16

Reject